# Text-Guided Attention is All You Need for Zero-Shot Robustness in Vision-Language Models

Lu Yu [1]     Haiyang Zhang [1]     Changsheng Xu [2]

[1]School of Computer Science and Engineering, Tianjin University of Technology
[2]State Key Laboratory of Multimodal Artificial Intelligence Systems,
Institute of Automation, University of Chinese Academy of Sciences
`{luyu@email, zshy@stud}.tjut.edu.cn, csxu@nlpr.ia.ac.cn`

## Abstract

Due to the impressive zero-shot capabilities, pre-trained vision-language models (e.g. CLIP), have attracted widespread attention and adoption across various domains. Nonetheless, CLIP has been observed to be susceptible to adversarial examples. Through experimental analysis, we have observed a phenomenon wherein adversarial perturbations induce shifts in text-guided attention. Building upon this observation, we propose a simple yet effective strategy: *Text-Guided Attention for Zero-Shot Robustness (TGA-ZSR)*. This framework incorporates two components: the Attention Refinement module and the Attention-based Model Constraint module. Our goal is to maintain the generalization of the CLIP model and enhance its adversarial robustness: The Attention Refinement module aligns the text-guided attention obtained from the target model via adversarial examples with the text-guided attention acquired from the original model via clean examples. This alignment enhances the model's robustness. Additionally, the Attention-based Model Constraint module acquires text-guided attention from both the target and original models using clean examples. Its objective is to maintain model performance on clean samples while enhancing overall robustness. The experiments validate that our method yields a 9.58% enhancement in zero-shot robust accuracy over the current state-of-the-art techniques across 16 datasets. *Our code is available at https://github.com/zhyblue424/TGA-ZSR.*

## 1   Introduction

Large-scale pre-trained vision-language models (VLMs) have showcased remarkable success in artificial intelligence by seamlessly integrating visual and textual data to understand complex multimodal information, such as CLIP [48]. Leveraging vast datasets and powerful architectures such as BERT [10] and its variants [8, 33], these models adeptly capture semantic relationships between images and texts, offering significant advantages across numerous applications. From image classification [14, 67, 55] and semantic segmentation [50] to image captioning [39] and vision question answering [44], pre-trained VLMs revolutionize how machines perceive and interact with multimodal information. Their importance lies in their ability to learn rich representations from varied data streams, enabling zero-shot learning and transfer learning across domains and tasks. Thus ensuring the reliability of large-scale models is crucial. However, these models are vulnerable to adversarial attacks as many other networks as demonstrated by recent studies [38, 59], even slight perturbations to input data can result in misclassification or altered outputs. Such attacks pose a significant challenge, particularly in critical applications like autonomous vehicles [60], medical diagnosis [32], and maritime navigation [29], where the consequences of erroneous decisions can be severe. As these large-scale models become increasingly prevalent in real-world applications, understanding and

38th Conference on Neural Information Processing Systems (NeurIPS 2024).

mitigating the risks posed by adversarial attacks is essential to maintain trust and reliability in AI systems.

Adversarial training [53, 61, 69] has emerged as a crucial technique in enhancing the robustness of deep learning models against adversarial attacks. By augmenting training data with adversarial examples generated through perturbations of input data, models are forced to learn more robust decision boundaries, thereby improving their resilience to adversarial manipulation. Given the rising significance of large-scale VLMs in various applications, understanding their vulnerability to adversarial attacks is essential. While adversarial training presents practical challenges when applied to downstream tasks, especially with large-scale models. Firstly, adversarial training typically involves generating adversarial examples during each training iteration, which increases the computational overhead and may lead to overfitting on the training data. This phenomenon is exacerbated in large-scale models with vast parameter spaces, where fine-tuning becomes more susceptible to overfitting. Moreover, adversarial training may not adequately prepare models for all possible adversarial scenarios, potentially leaving them vulnerable to unknown data distributions encountered in real-world settings. Exploring zero-shot adversarial robustness in these models is particularly pertinent as it sheds light on their ability to generalize and perform reliably in unseen scenarios. Additionally, considering the multimodal nature of VLMs, the exploration of zero-shot adversarial robustness offers insights into the complex interactions between visual and textual modalities, paving the way for more robust and trustworthy multimodal AI systems.

Text-guided Contrastive Adversarial Training (TeCoA) method [38] represents the pioneering effort in investigating the zero-shot adversarial robustness of large-scale VLMs. They aim to bolster CLIP's zero-shot generalization capacity against adversarial inputs. While their primary focus lies on enhancing accuracy in the face of adversarial samples, this improvement comes at the expense of decreased performance on clean data. Subsequent work by PMG-AFT [59] builds upon this by introducing a pre-trained model guided adversarial fine-tuning technique, further enhancing both generalizability and adversarial robustness. However, despite the advancements made by both studies in enhancing CLIP's zero-shot robustness, significant questions regarding the interpretability of adversarial attacks and the efficacy of adversarial training remain unanswered. Specifically, *the mechanisms through which adversarial attacks influence network outputs and the reasons behind the effectiveness of adversarial training strategies remain elusive*. In our paper, we delve into the text-guided attention shift phenomenon to shed light on how adversarial attacks alter model outputs. Leveraging these insights, we propose a simple yet effective strategy, TGA-ZSR, aimed at enhancing the robustness of the CLIP model and preserving its performance on clean examples.

Our main contributions are summarized follows:

- To our knowledge, we are the first to introduce text-guided attention to enhance zero-shot robustness on vision-language models while maintaining performance on clean sample.
- We improve the interpretability of adversarial attacks for zero-shot robustness on vision-language models through a text-guided attention mechanism.
- The experimental results show that TGA-ZSR surpasses previous state-of-the-art methods, establishing a new benchmark in model zero-shot robust accuracy.

## 2 Related Work

**Pre-trained Vision-language Models.** In recent years, advancements in computer vision[12, 17, 34] have primarily relied on training models with image-label pairs to recognize predefined object categories. However, these approaches often overlook the inherent semantic connections between textual descriptions and visual content. Motivated by the remarkable progress witnessed in natural language processing (NLP), exemplified by breakthroughs like Transformer [56], BERT [10], and GPT-3 [3], researchers are increasingly drawn to the prospect of using textual data to enhance the capabilities of DNNs. These methodologies are referred to as VLMs [21, 48, 49, 64] and one prominent approach is to directly learn the semantic similarity between images and corresponding textual descriptions through image-text pairs. By aligning the embeddings of these two modalities, models like CLIP [48], ALIGN [21], BLIP [25], Visual-BERT [47], and ALBEF [26] aim to achieve superior performance across various tasks. CLIP [48] leverages a vast dataset of 400 million image-text pairs sourced from the internet and employs contrastive loss to effectively align the embeddings of both modalities, thereby enhancing the model's capabilities. Experimental results underscore the

significant performance gains achieved by incorporating textual information into the model, with zero-shot performance surpassing that of earlier deep neural network architectures. However, despite its impressive zero-shot accuracy, experiments [38, 59] reveal vulnerabilities to adversarial examples, resulting in a notable decline in robustness.

**Adversarial Robustness.** Deep neural networks have been found to be vulnerable to adversarial examples [54, 36, 40, 66], which can fool DNNs to produce false outputs, rendering trained models unreliable. To bolster robustness against such adversarial attacks, various advanced methods have been proposed, including data augmentation [28, 58, 27, 65], adversarial training [69, 53, 61, 68], progressive self-distillation [1], randomization strategy [11, 35], and adversarial purification [41, 24, 62]. While these strategies aim to improve DNNs' adversarial robustness, they often come with increased complexity or limited generalizability. Adversarial training [69, 53, 61, 68] stands out as one of the most widely used and effective approaches, fine-tuning DNNs by generating adversarial examples during training. After the emergence of CLIP [48], many subsequent works [45, 16, 63] have utilized CLIP as a backbone, yet little attention has been given to studying its adversarial robustness. CLIP is shown to be susceptible to adversarial examples [38] as well, posing a significant threat to downstream tasks utilizing CLIP as a backbone. Hence, investigating the adversarial robustness of CLIP is crucial.

**Zero-shot Adversarial Robustness for VLMs.** The visual-language model, trained on both image and text data, serves as a foundational model for various tasks. However, it has shown vulnerability to adversarial examples [38, 59], and training from scratch is time-intensive. TeCoA [38] was the first to explore zero-shot adversarial robustness for VLMs, aiming to enhance CLIP's adversarial robustness by minimizing the cross-entropy loss between image logits and targets. While TeCoA solely utilizes cross-entropy loss, yielding only marginal performance improvements, PMG-AFT [59] extends this approach by minimizing the distance between features of adversarial examples and those of the pre-trained model. FARE [51] primarily focuses on maintaining high clean accuracy while improving model robustness, achieving this by constraining the distance between the original and target model embeddings. Our experiments reveal significant differences in attention maps between original examples and adversarial examples. Leveraging this insight, we enhance model robustness by constraining it with text-guided attention.

## 3 Methodology

### 3.1 Preliminaries and Problem Setup

Following the previous works [38, 59], we choose CLIP model as the pre-trained VLMs for image classification task. Given an image-text pair $(x, t)$, where $x$ represents an image and $t$ represents a textual prompt, CLIP learns to encode both the image and the text into fixed-dimensional embeddings. Let $f(x)$ denote the embedding of the image $x$ and $g(t)$ denote the embedding of the text prompt $t$, $y$ is the one-hot vector label. For training or fine-tuning on the downstream tasks, we use the cross-entropy loss, denoted as $L(x, t, y)$.

$$L(x, t, y) = -\mathbb{E}_{i,j}\left[y_{ij} log \frac{exp(cos(f(x)_i, g(t)_j))/\tau)}{\sum_k exp(cos(f(x)_i, g(t)_k))/\tau)}\right] \tag{1}$$

where we set $y_{ij} = 1$ if the image-text pair is positive, otherwise, $y_{ij} = 0$. $\tau$ is the temperature parameter and $cos$ indicates calculating the cosine similarity of the two embeddings.

**Adversarial Attacks.** Adversarial attacks are a concerning phenomenon where small, often imperceptible perturbations are intentionally applied to input data with the aim of deceiving a model into producing incorrect outputs. These perturbations are crafted with the goal of causing the model to misclassify or generate erroneous predictions while appearing indistinguishable to human observers. The Projected Gradient Descent (PGD) [36] method is an iterative approach for crafting adversarial examples. It starts with the original input data and then iteratively adjusts the data in the direction that maximizes the model's loss function while ensuring the perturbed data remains within a specified perturbation budget. Mathematically, the PGD attack can be expressed as follows:

$$x_{a+1} = \Pi_{x+S}(x_a + \varepsilon \cdot sign(\bigtriangledown_{x_a} L(x_a, t, y))) \tag{2}$$

Here, $L$ represents the loss function, $x$ denotes the original input data, $\varepsilon$ controls the magnitude of perturbation, and $\bigtriangledown_x L$ represents the gradient of the loss function with respect to the input data.

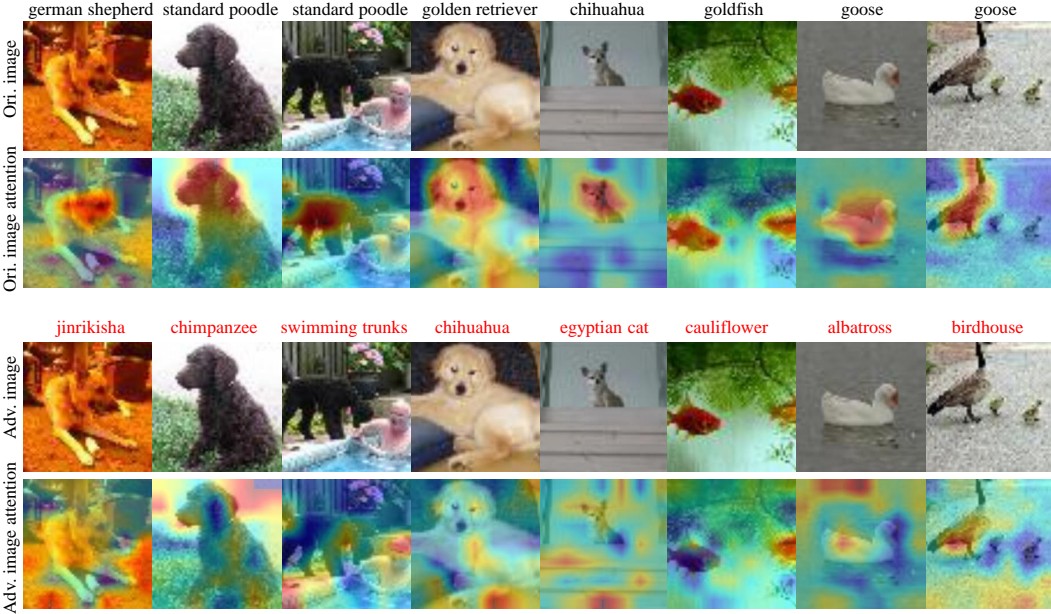

Figure 1: The four rows depict the original image, its associated attention map, the generated adversarial example, and the attention map of the adversarial example. Labels in black indicate the ground truth, while those in red represent mis-classified labels for the adversarial examples.

By adding or subtracting $\varepsilon$ times the sign of this gradient to the original input data, the PGD attack generates adversarial examples that lead to misclassification or incorrect predictions by the model. $\Pi_{x+S}$ makes the perturbed data remains within an $\varepsilon$-neighborhood of the original input, preventing the generated adversarial examples from straying too far. $S$ is a set of allowed perturbations that formalizes the manipulative power of the adversary.

**Adversarial Examples Generation and Adversarial Training.** The optimization objective for crafting adversarial examples aims to maximize the loss of model $f_\theta$ with respect to a perturbed input $x_a$ which can be formulated as:

$$x_a = \underset{x_a}{argmax}\, L(f_\theta(x_a, t, y)) \tag{3}$$

Adversarial training is a technique to generate adversarial examples from the original training data and then use these examples to train the model, forcing it to learn to resist adversarial perturbations. To adapt the model to the downstream tasks, we apply adversarial fine-tuning on one target model towards robustness with the following loss:

$$\theta = \underset{\theta}{argmin}\, \mathcal{J}(f_\theta(x_a, t, y)) \tag{4}$$

Where $\mathcal{J}$ represents the total loss function used for training the model.

**Zero-Shot Adversarial Robustness.** In this paper, we investigate the zero-shot adversarial robustness of CLIP model, which refers to the ability of these models to maintain performance and reliability even when encountering unseen adversarial samples during inference, with only adversarial fine-tuning the original CLIP model on one target dataset, such as Tiny-ImageNet.

## 3.2 Text-Guided Attention based Interpretation of Adversarial Attacks

**Text-Guided Attention.** Attention mechanisms [30, 16, 31] play a crucial role in enhancing vision model performance across various tasks. At its core, attention enables models to focus on relevant parts of the input data while suppressing irrelevant information. Similarly, in VLMs, by incorporating textual guidance, the models can effectively focus on relevant visual features while processing

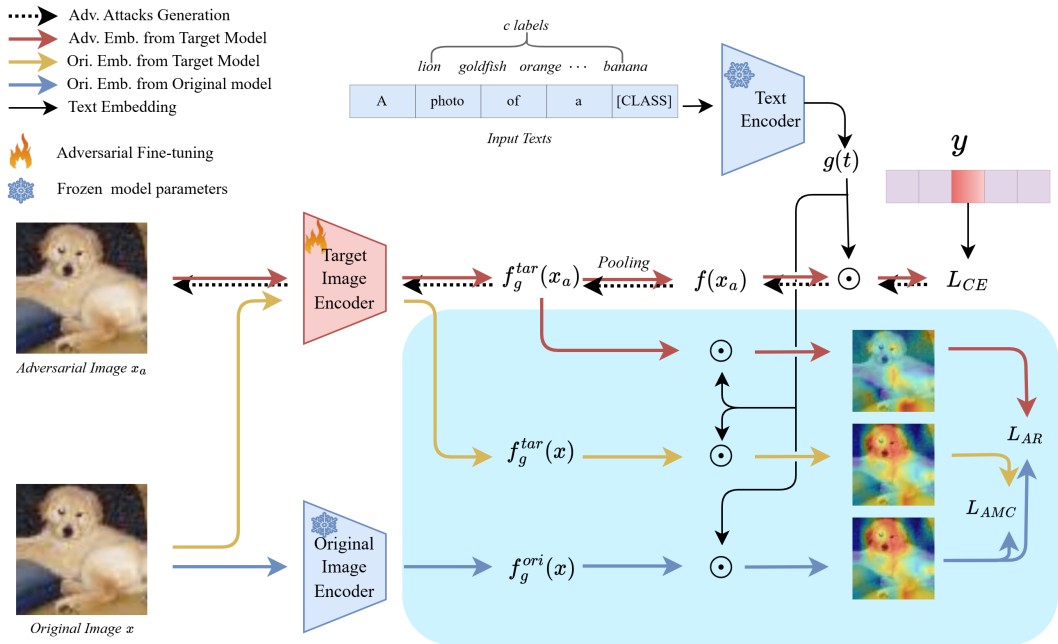

Figure 2: An overview of our TGA-ZSR framework: We generate adversarial examples and feed them into the target image encoder. To enhance the adversarial robustness of the CLIP model and maintain its generalization, we introduce text-guided attention. This involves refining the framework for adversarial examples through the Attention Refinement module and constraining the model to prevent significant drift via the Attention-based Model Constraint module.

language, thus facilitating more accurate and coherent multimodal understanding. Additionally, text-guided attention enhances interpretability by providing insights into the model's decision-making process, fostering trust and understanding in complex multimodal systems. Thus, we investigate the impact of text-guided attention on enhancing and interpreting zero-shot adversarial robustness in VLMs in this paper. We define the text-guided attention as following:

$$A(x) = f_g(x) \cdot g(t)^{\mathsf{T}}, \quad A \in \mathbb{R}^{P \times 1} \tag{5}$$

Where $f_g(x)$ represents the global image feature before the pooling operation of $f(x)$, and $P$ denotes the dimension of the attention embeddings. We reshape $A$ to $\mathbb{R}^{\sqrt{P} \times \sqrt{P}}$ to obtain the attention map, which is then resized to $A \in \mathbb{R}^{H \times W}$. Finally, we apply a normalization operation ($norm$) on $A$ to obtain the final text-guided attention map.

**Interpretation of Adversarial Attacks.** The previous research has predominantly focused on bolstering the zero-shot robustness of Vision-Language Models (VLMs), yet the reasons leading to mis-classifications induced by adversarial attacks remain unclear. This paper aims to shed light on interpreting the impact of adversarial attacks on VLMs. By employing Eq. 5, we compute the text-guided attention for both the original image (*Ori. image*) and its corresponding adversarial counterpart (*Adv. image*), as depicted in Fig. 1. Remarkably, despite the subtle discrepancies imperceptible to the human eye between the adversarial example and the original image, the former is mis-classified (labels in red). However, a significant difference emerges in the respective text-guided attention maps. Specifically, we observe a notable shift in the text-guided attention of the adversarial example, characterized by instances of displacement towards other objects, backgrounds, or even disappearance. For instance, while the original images in the first, second, and fourth columns pay attention to their subjects' heads, in their adversarial counterparts, attention diverges elsewhere. In the third column, the attention shift leads from the correct object to an incorrect one, resulting in mis-classification. In the fifth and seventh columns, the attention in their adversarial counterparts is redirected towards the background.

### 3.3 Text-Guided Attention for Zero-Shot Robustness (TGA-ZSR)

The semantic information embedded within text representations are preserved through a frozen text encoder, offering invaluable guidance when adversarial perturbations disrupt relevant visual features, which has not been explored for zero-shot robustness of vision-language models. We introduce the Attention Refinement Module, designed to effectively filter out irrelevant information, thereby mitigating the impact of adversarial attacks seeking to exploit vulnerabilities in the model's decision-making process. Moreover, to maintain model's ability to generalize effectively on clean images, we introduce the Attention-based Model Constraint Module. This module ensures consistent performance on clean data while enhancing the model against adversarial disruptions. Additionally, employing text-guided attention enhances interpretability, offering crucial insights into how the model integrates and processes information across modalities. This interpretability not only instills trust in the model's predictions but also facilitates the detection and mitigation of adversarial attacks. Our approach (i.e. TGA-ZSR) presents a comprehensive framework (as shown in Fig. 2) for enhancing model robustness to adversarial perturbations while concurrently improving interpretability. We will introduce the details as follows.

**Attention Refinement Module.** Based on the insights gained in Section 3.2, we propose an attention refinement module aimed at enhancing the robustness of the model. This module is designed to rectify the text-guided attention of adversarial samples, which often leads to altered predictions. Our approach aligns the adversarial attention map with that of the clean samples, known for their high-accuracy attention distribution. This simple yet effective strategy serves to mitigate the impact of adversarial perturbations on the model's predictions.

We take the generated adversarial sample $x_a$ to the target model $f_g^{tar}(\cdot)$ and the clean sample $x$ to the original model $f_g^{ori}(\cdot)$ and obtain the adversarial attention map $A(x_a^i)_{tar}$ and the clean attention map $A(x^i)_{ori}$ respectively. The attention refinement loss $L_{AR}$ is thus defined as:

$$L_{AR} = \frac{1}{N} \cdot \sum_{i=0}^{N} \|A(x_a^i)_{tar} - A(x^i)_{ori}\|_2 \tag{6}$$

where $A(x_a)_{tar} = f_g^{tar}(x_a) \cdot g(t)^\mathsf{T}$ and $A(x)_{ori} = f_g^{ori}(x) \cdot g(t)^{\mathsf{T}}$ [1], $\|\|_2$ denotes the $L_2$ distance computation between two attention maps.

**Attention-based Model Constraint Module.** The Attention Refinement module serves to enhance the robustness of the models, consequently improving the accuracy of adversarial samples. However, this enhancement comes with a trade-off: it may marginally sacrifice the accuracy on clean samples due to shifts in model parameters. To preserve the generalization capability of pre-trained VLMs, we introduce an Attention-based Model Constraint module. This module aims to mitigate performance drops on clean images, thereby ensuring the overall effectiveness and reliability of the model.

Specifically, we input the clean sample $x$ into the target model $f_g^{tar}(\cdot)$, adversarially fine-tuned on the Tiny-ImageNet dataset, to acquire the text-guided attention map $A(x)_{tar}$. Concurrently, the original text-guided attention map outputted from the original CLIP model $f_g^{ori}(\cdot)$ is denoted as $A(x)_{ori}$. To ensure the preservation of importance parameters for clean images, we enforce an $L_2$ distance constraint between these two attention maps. The attention-based model constraint loss $L_{AMC}$ is formulated as:

$$L_{AMC} = \frac{1}{N} \cdot \sum_{i=0}^{N} \left\|A(x^i)_{tar} - A(x^i)_{ori}\right\|_2 \tag{7}$$

Thus the final loss function can be represented as:

$$L_{total} = L_{CE} + \alpha \cdot L_{AR} + \beta \cdot L_{AMC} \tag{8}$$

## 4 Experiments

### 4.1 Experimental Setup

**Datasets.** Our experiments begin with training the pre-trained CLIP model on the Tiny-ImageNet [9]. Then we evaluate the model's zero-shot adversarial robustness across 15 subsequent datasets, fol-

---

[1]We only compute the attention map for the image corresponding to the text prompt of the ground-truth label.

Table 1: Zero-shot robust accuracy on images attacked with 100 steps of PGD [36]. We performed several different methods on Tiny-ImageNet and evaluated across 16 datasets. The optimal accuracy is highlighted in **bold**, while the second-best accuracy is underlined. The values in parentheses represent the standard deviation.

| Methods | Tiny-ImageNet | CIFAR-10 | CIFAR-100 | STL-10 | SUN397 | Food101 | OxfordPets | Flowers102 | DTD | EuroSAT | FGVC-Aircraft | ImageNet | Caltech-101 | Caltech-256 | StanfordCars | PCAM | Average |
|---|---|---|---|---|---|---|---|---|---|---|---|---|---|---|---|---|---|
| CLIP [48] | 0.88 | 2.42 | 0.26 | 26.11 | 1.00 | 6.60 | 3.84 | 1.19 | 2.02 | 0.05 | 0.00 | 1.24 | 19.88 | 12.60 | 0.20 | 0.11 | 4.90 |
| FT-Clean | 13.55 | 19.92 | 4.94 | 40.00 | 0.82 | 0.64 | 2.40 | 0.68 | 2.66 | 0.05 | 0.03 | 1.08 | 14.95 | 9.69 | 0.09 | 1.32 | 7.05 |
| FT-Adv. | 51.59 | 38.58 | 21.28 | 69.55 | 17.60 | 12.55 | 34.97 | 19.92 | 15.90 | 11.95 | 1.83 | 17.26 | 50.73 | 40.18 | 8.42 | 48.88 | 28.83 |
| TeCoA [38] | 37.57 | 30.30 | 17.53 | 67.19 | 19.70 | 14.76 | 36.44 | 22.46 | 17.45 | 12.14 | 1.62 | 18.18 | 55.86 | 41.88 | 8.49 | 47.39 | 28.06 |
| FARE[51] | 23.88 | 21.25 | 10.72 | 59.59 | 8.30 | 10.97 | 24.56 | 15.48 | 10.96 | 0.14 | 0.84 | 10.54 | 45.96 | 34.35 | 4.38 | 10.17 | 18.25 |
| PMG-AFT[59] | 47.11 | 46.01 | 25.83 | 74.51 | 22.21 | 19.58 | 41.62 | 23.45 | 15.05 | 12.54 | 1.98 | 21.43 | 62.42 | 45.99 | 11.72 | 48.64 | 32.51 |
| TGA-ZSR (ours) | **63.95** | **61.45** | **35.27** | **84.22** | **33.22** | **33.97** | **57.75** | **34.55** | **22.08** | **14.27** | **4.75** | **28.74** | **70.97** | **60.06** | **20.40** | 47.76 | **42.09** |
| | (± 0.11) | (± 0.67) | (± 0.07) | (± 0.21) | (± 0.39) | (± 0.20) | (± 0.76) | (± 0.35) | (± 0.16) | (± 0.26) | (± 0.27) | (± 0.11) | (± 0.42) | (± 0.46) | (± 0.68) | (± 0.35) | (± 0.12) |

Table 2: Zero-shot clean accuracy. We performed several different methods on Tiny-ImageNet and evaluated across 16 datasets. The values in parentheses represent the standard deviation.

| Methods | Tiny-ImageNet | CIFAR-10 | CIFAR-100 | STL-10 | SUN397 | Food101 | OxfordPets | Flowers102 | DTD | EuroSAT | FGVC-Aircraft | ImageNet | Caltech-101 | Caltech-256 | StanfordCars | PCAM | Average |
|---|---|---|---|---|---|---|---|---|---|---|---|---|---|---|---|---|---|
| CLIP [48] | 57.26 | **88.06** | 60.45 | **97.04** | 57.26 | **83.89** | 87.41 | 65.47 | 40.69 | 42.59 | 20.25 | 59.15 | 85.34 | 81.73 | 52.02 | 52.09 | 64.42 |
| FT-Clean | 79.04 | 84.55 | 54.25 | 93.78 | 46.80 | 47.10 | 80.98 | 46.43 | 30.32 | 24.39 | 9.30 | 44.40 | 78.69 | 70.81 | 31.15 | 47.89 | 54.37 |
| FT-Adv. | 73.83 | 68.96 | 39.69 | 86.89 | 33.37 | 27.74 | 60.10 | 33.45 | 23.14 | 16.49 | 4.86 | 32.06 | 67.41 | 57.72 | 18.11 | 49.91 | 43.36 |
| TeCoA [38] | 63.97 | 66.14 | 36.74 | 87.24 | 40.54 | 35.11 | 66.15 | 38.75 | 25.53 | 17.13 | 6.75 | 37.09 | 74.63 | 62.50 | 24.65 | 50.01 | 45.81 |
| FARE[51] | 77.54 | 87.58 | **62.80** | 94.33 | 49.91 | 70.02 | 81.47 | 57.10 | 36.33 | 22.69 | 14.19 | 51.78 | 84.04 | 77.50 | 44.35 | 46.07 | 59.85 |
| PMG-AFT[59] | 67.11 | 74.62 | 44.68 | 88.85 | 37.42 | 37.47 | 66.34 | 35.66 | 21.17 | 17.76 | 4.71 | 35.93 | 76.70 | 61.96 | 25.21 | 49.99 | 46.60 |
| TGA-ZSR(ours) | 75.72 | 86.46 | 56.52 | 93.48 | 51.99 | 57.59 | 77.32 | 48.08 | 29.06 | 24.24 | 11.93 | 48.04 | 80.70 | 74.74 | 36.62 | 49.58 | 56.44 |
| | (± 0.12) | (± 0.26) | (± 0.35) | (± 0.19) | (± 0.25) | (± 0.34) | (± 0.30) | (± 0.37) | (± 0.35) | (± 0.49) | (± 0.27) | (± 0.06) | (± 0.09) | (± 0.18) | (± 1.03) | (± 0.17) | (± 0.08) |

lowed by previous studies, such as TeCoA [38] and PMG-AFT [59]. These datasets include several commonly used classfication datasets, including CIFAR-10 [23], CIFAR-100 [23], STL-10 [6], ImageNet [9], Caltech-101 [13], and Caltech-256 [15]. Additionally, fine-grained image classification datasets such as StanfordCars [22], Flowers102 [42], Food101 [2], FGVCAircraft [37], and Oxford-Pets [46] are included. Furthermore, the scene recognition dataset SUN397 [43], the medical image dataset PCAM [57], and the satellite image classifacation dataset EuroSAT [18] and the texture recognition dataset DTD [5] are incorporated for comprehensive evaluation. We also conduct experiments on four additional datastes (i.e. ImageNet_subset, ImageNet-A, ImageNet-O and ImageNet-R) as shown in Supp. Mat. A.1.

**Implementation Details.** Following the protocol of previous works [59], we fine-tuned the CLIP model on the adversarial samples of Tiny-ImageNet [9] as *'adversarial fine-tuning'* and subsequently evaluated its performance across 15 datasets and Tiny-ImageNet itself. We employ ViT-B/32 as the backbone in CLIP and utilize the SGD optimizer to minimize loss. During adversarial fine-tuning, we update all parameters of the image encoder with a learning rate of 1e-4, weight decay of 0, momentum of 0.9, and a batch size of 128. We utilize $l_\infty$ norm PGD-2 [36] with 2 iterations to generate adversarial examples, with an attack strength $\varepsilon$ of 1/255 and the attack step size is 1/255. To evaluate zero-shot adversarial inference, we employ $l_\infty$ norm PGD-100 [36] with 100 iterations, attack step of 1/255 and a batch size of 256 to generate adversarial examples for verifying CLIP's adversarial robustness. Additionally, to assess the model's robustness under different attack strengths, we perform inference using adversarial strengths $\varepsilon$ of 1/255, 2/255, and 4/255. The hyper-parameters $\alpha$ and $\beta$ are set to 0.08 and 0.05 respectively in Eq. 8 in the main experiments. Maintain the same parameters for the CW attack. For the AutoAttack [7] experiments, $\alpha$ and $\beta$ are set to 0.08 and 0.009. We conducted the experiment utilizing the RTX 3090, which required a training period ranging from 3 to 4 hours.

## 4.2 Main Results

To validate the effectiveness of our approach, we conduct comparisons with several state-of-the-art methods such as TeCoA [38], PMG-AFT [59], and FARE [51]. Additionally, we extend the comparison to include CLIP (the original pre-trained CLIP model), FT-Adv. (adversarial fine-tuning using the contrastive loss of the original CLIP) and FT-Clean (fine-tuning on clean examples with the contrastive loss of the original CLIP) for a comprehensive evaluation.

Table 3: Zero-shot robust accuracy on images attacked with $\varepsilon$ of 1/255 of AutoAttack [7]. We performed several different methods on Tiny-ImageNet and evaluated on 16 datasets.

| Methods | Tiny-ImageNet | CIFAR-10 | CIFAR-100 | STL-10 | SUN397 | Food101 | Oxfordpets | Flowers102 | DTD | EuroSAT | FGVC-Aircraft | ImageNet | Caltech-101 | Caltech-256 | StanfordCars | PCAM | Average |
|---|---|---|---|---|---|---|---|---|---|---|---|---|---|---|---|---|---|
| CLIP [48] | 0.02 | 0.01 | 0.08 | 0.03 | 0.04 | 0.01 | 0.00 | 0.03 | 0.16 | 0.12 | 0.06 | 0.04 | 0.43 | 0.10 | 0.11 | 0.22 | 0.09 |
| FT-Clean | 0.08 | 0.03 | 0.01 | 0.91 | 0.09 | 0.04 | 0.06 | 0.03 | 0.48 | 0.02 | 0.03 | 0.12 | 1.38 | 0.66 | 0.03 | 0.03 | 0.25 |
| FT-Adv. | **50.48** | 37.55 | 20.39 | 69.14 | 16.25 | 11.23 | 33.91 | 18.54 | **19.95** | 11.59 | 1.65 | 16.21 | 49.90 | 39.24 | 7.57 | **48.84** | 28.28 |
| TeCoA [38] | 35.03 | 28.18 | 16.09 | 66.08 | 17.41 | 13.05 | 34.81 | 20.80 | 15.37 | 11.40 | 1.32 | 16.32 | 54.54 | 40.15 | 7.15 | 47.12 | 26.55 |
| FARE [51] | 28.59 | 23.37 | 13.58 | 60.70 | 9.72 | 13.88 | 27.72 | 15.48 | 9.15 | 0.25 | 0.87 | 12.07 | 47.45 | 36.68 | 6.77 | 10.23 | 19.78 |
| PMG-AFT [59] | 44.26 | **44.12** | **23.66** | **73.90** | 19.63 | **17.25** | 39.25 | 20.87 | 13.72 | **11.99** | 1.68 | 19.17 | **60.57** | 44.25 | 9.59 | 48.53 | 30.78 |
| TGA-ZSR (ours) | 49.45 | 40.53 | 22.38 | 72.06 | **20.36** | 15.58 | **40.31** | **21.43** | 17.13 | 11.19 | **2.64** | **19.28** | 57.16 | **45.68** | **10.47** | 48.03 | **30.86** |

Table 4: Zero-shot robust accuracy across 16 datasets with CW attack [4]. The optimal accuracy is highlighted in **bold**.

| Methods | Tiny-ImageNet | CIFAR-10 | CIFAR-100 | STL-10 | SUN397 | Food101 | Oxfordpets | Flowers102 | DTD | EuroSAT | FGVC-Aircraft | ImageNet | Caltech-101 | Caltech-256 | StanfordCars | PCAM | Average |
|---|---|---|---|---|---|---|---|---|---|---|---|---|---|---|---|---|---|
| CLIP [48] | 0.21 | 0.36 | 0.10 | 10.59 | 1.16 | 0.82 | 1.23 | 1.09 | 2.18 | 0.01 | 0.00 | 1.14 | 13.50 | 7.36 | 2.36 | 0.07 | 3.64 |
| PMG-AFT[59] | 44.59 | 44.86 | 24.15 | 74.11 | 19.99 | 17.33 | 39.88 | 20.95 | 13.51 | 12.09 | 1.47 | 19.51 | 60.99 | 44.46 | 10.57 | **48.59** | 31.07 |
| TGA-ZSR(ours) | **63.85** | **60.50** | **34.62** | **84.11** | **22.03** | **33.28** | **58.33** | **32.95** | **21.22** | **13.89** | **4.56** | **20.42** | **70.34** | **59.73** | **20.20** | 48.02 | **40.50** |

**Adversarial Zero-shot Robust Accuracy.** Table 1 shows that the average accuracy of our TGA-ZSR outperforms the original CLIP model by 37.19%. Compared to current stat-of-the-art method, PMG-AFT, the proposed method achieve an average improvement of 9.58%. In general, our method is superior than all the other methods on most datasets except a comparable result on PCAM dataset. In addition, we obtain the best result on Tiny-ImageNet, which is not a strict zero-shot test. It indicates that our method is robust on the adversarial attack on both seen and unseen datasets.

**Zero-shot Clean Accuracy.** Table 2 illustrates the model's accuracy for clean examples using different methods. Our method outperforms PMG-AFT by 9.84% and FT-clean by 2.07% in terms of average accuracy. Similar to Table 1, zero-shot clean accuracy exhibits improvement not only on an individual dataset but across all datasets. However, we observed that our zero-shot clean accuracy is 3.41% lower than that achieved by FARE. It is important to note that FARE prioritizes preserving zero-shot clean accuracy. However, we have significantly enhanced the zero-shot robust accuracy in adversarial scenarios with 23.84% gain compared to FARE in Table 1.

### 4.3 Experiments on More Attack Types

**Results against AutoAttack.** AutoAttack [7] stands out as a strong attack method for assessing model robustness. We follow TeCoA and PMG-AFT to verify the perturbation bound $\varepsilon$ of 1/255 in the standard version of AutoAttack. The results are summarized in Table 3. We can see that the original CLIP model experienced a significant performance decline, decreasing to 0.09% on the adversarial example. Our TGA-ZSR also demonstrates a decline but still achieves superior results compared to other methods, validating its effectiveness against stronger attacks.

**Results against CW Attack.** CW attack [4] is an optimization-based approach designed to generate small perturbations to input data, causing the model to make incorrect predictions while keeping the perturbed input visually similar to the original. We further evaluate the robustness of our approach against this challenging attack, using a perturbation bound of $\varepsilon = 1/255$. The results, shown in Table 4, demonstrate that our method significantly outperforms the state-of-the-art method PMG-AFT on both adversarial and clean samples. This substantial margin indicates the adversarial robustness of our proposed method.

### 4.4 Ablation Study

**Different Types of Attentions.** To validate the important role of text-guided attention in our method, we conducted experiments by replacing it with vision-based attention. We employ Grad-CAM [52], a widely adopted method, for generating attention maps based on vision. Table 5 demonstrates that replacing the text-guided attention with vision-based attention yields results that are still comparable to

Table 5: Comparison of vision-based attention and our text-guided attention. We evaluate the state-of-the-art method PMG-AFT alongside our pipeline, incorporating two different types of attention mechanisms on Tiny-ImageNet and evaluating performance across 16 datasets.

| Test | Methods | Tiny-ImageNet | CIFAR-10 | CIFAR-100 | STL-10 | SUN397 | Food101 | OxfordPets | Flowers102 | DTD | EuroSAT | FGVC-Aircraft | ImageNet | Caltech-101 | Caltech-256 | StanfordCars | PCAM | Average |
|---|---|---|---|---|---|---|---|---|---|---|---|---|---|---|---|---|---|---|
| Robust | PMG-AFT[59] | 47.11 | 46.01 | 25.83 | 74.51 | 22.21 | 19.58 | 41.62 | 23.45 | 15.05 | 12.54 | 1.98 | 21.43 | 62.42 | 45.99 | 11.72 | **48.64** | 32.51 |
| | Vision-based | 52.81 | 40.46 | 22.66 | 70.26 | 19.50 | 13.74 | 37.67 | 19.78 | 16.97 | 11.79 | 2.64 | 18.08 | 55.64 | 42.45 | 8.88 | 38.11 | 29.47 |
| | TGA-ZSR (ours) | **63.97** | **61.82** | **35.25** | **83.99** | **32.78** | **34.13** | **56.91** | **34.20** | **21.92** | **14.20** | **4.44** | **28.62** | **70.53** | **59.70** | **21.15** | 47.75 | **41.96** |
| Clean | PMG-AFT[59] | 67.11 | 74.62 | 44.68 | 88.85 | 37.42 | 37.47 | 66.34 | 35.66 | 21.17 | 17.76 | 4.71 | 35.93 | 76.70 | 61.96 | 25.21 | 49.99 | 46.60 |
| | Vision-based | 74.31 | 70.77 | 41.03 | 87.24 | 36.91 | 30.07 | 62.52 | 33.89 | 24.10 | 16.26 | 5.70 | 33.59 | 72.35 | 59.75 | 20.50 | **51.29** | 45.02 |
| | TGA-ZSR (ours) | **76.85** | **86.23** | **56.55** | **93.28** | **51.71** | **57.72** | **77.08** | **48.32** | **29.15** | **23.99** | **12.03** | **48.10** | **80.82** | **74.58** | **37.72** | 49.60 | **56.48** |

Table 6: Zero-shot robust accuracy on images attacked with $\varepsilon$ of 1/255, 2/255 and 4/255 of PGD [36]. We performed several different methods on Tiny-ImageNet and evaluated across 16 datasets. We represent the **average** accuracy across various attack strength.

| Methods | Tiny-ImageNet | CIFAR-10 | CIFAR-100 | STL-10 | SUN397 | Food101 | OxfordPets | Flowers102 | DTD | EuroSAT | FGVC-Aircraft | ImageNet | Caltech-101 | Caltech-256 | StanfordCars | PCAM | Average |
|---|---|---|---|---|---|---|---|---|---|---|---|---|---|---|---|---|---|
| CLIP [48] | 0.64 | 2.15 | 0.12 | 20.35 | 0.52 | 5.94 | 2.97 | 0.72 | 0.71 | 0.03 | 0.00 | 0.71 | 14.28 | 9.18 | 0.11 | 0.04 | 3.65 |
| FT-Clean | 12.44 | 18.80 | 4.65 | 37.16 | 0.43 | 0.52 | 2.03 | 0.41 | 0.92 | 0.02 | 0.01 | 0.54 | 13.02 | 7.96 | 0.03 | 0.44 | 6.21 |
| FT-Adv. | 29.33 | 18.10 | 11.06 | 45.13 | 8.58 | 5.65 | 16.45 | 10.15 | 9.72 | 9.82 | 0.83 | 8.81 | 33.43 | 24.14 | 3.80 | **38.06** | 17.07 |
| TeCoA [38] | 18.17 | 12.78 | 8.12 | 39.87 | 8.90 | 6.53 | 16.61 | 11.04 | **10.07** | 9.88 | 0.63 | 8.43 | 34.94 | 23.92 | 3.45 | 33.20 | 15.41 |
| PMG-AFT [59] | 25.30 | 21.71 | 13.29 | 47.69 | 11.42 | 9.49 | 20.68 | 12.86 | 9.45 | **10.65** | 0.90 | 11.28 | **41.86** | 28.38 | 5.40 | 37.88 | 19.27 |
| FARE [51] | 12.41 | 9.09 | 4.23 | 33.72 | 2.98 | 4.75 | 9.67 | 5.52 | 4.26 | 0.05 | 0.28 | 3.90 | 23.97 | 16.95 | 1.48 | 3.43 | 8.54 |
| TGA-ZSR (ours) | **33.40** | **27.72** | **14.68** | **59.59** | **12.40** | **12.99** | **24.69** | **13.42** | 9.70 | 7.47 | **1.53** | **11.69** | 41.44 | **32.51** | **7.29** | 19.64 | **20.45** |

the state-of-the-art method PMG-AFT in terms of both zero-shot robust accuracy and clean accuracy. This finding validates the effectiveness of our method's pipeline. Furthermore, our text-guided attention significantly improves the average accuracy, demonstrating the advantage of incorporating textual guidance.

**Effect of Attack Strength.** We further assess the robustness of the pre-trained model by different levels of PGD-2 attacks. Specifically, we set $\varepsilon$ to values of 1/255, 2/255 and 4/255, progressively amplifying the magnitude of the adversarial perturbation. This allows us to investigate whether a model trained on weak adversarial examples exhibits robustness against stronger adversarial perturbations. In Table 6, we present the average results for three distinct levels of attack strength. Despite a general decline in the robustness of all methods, our approach still achieves superior results, outperforming PMG-AFT by 1.18%, and FARE by 11.91%.

**Effect of Each Component.** We conducted several experiments to thoroughly evaluate the effectiveness of each component of our method, as summarized in Table 7. Using $L_{CE}$ alone significantly enhances the model's robustness through standard adversarial training, but it also results in a notable decrease in clean accuracy compared to the original CLIP model. Applying our Attention Refinement module further improves the average zero-shot accuracy on both adversarial and clean samples. Finally, the Attention-based Model Constraint module dramatically boosts performance, increasing robustness by 10.25% and clean accuracy by 6.52%.

**Trade-off between Robust and Clean Accuracy.** Achieving the balance between robustness and clean accuracy is crucial in adversarial training. Overfitting in models tends to yield high robustness but low clean accuracy, whereas underfitting typically results in the opposite scenario. As shown in Fig. 3, methods positioned close to the dotted line excel in either adversarial accuracy or clean accuracy, yet they often struggle to strike a balance between robustness and clean accuracy. In contrast, our method demonstrates not only an enhancement in the model's robustness but also the maintenance of clean accuracy, resulting in an overall superior performance.

More comprehensive and detailed ablation studies, including hyperparameter selection, the effect of distance metrics on the loss function, and the effect of learning rate, can be found in Supp. Mat. A.2.

## 4.5 Computational Overhead and Time Efficiency

We have evaluated our method against others in terms of memory usage, training time, and test time, and the results are summarized in Table 8. Our method increases memory consumption by approxi-

Table 7: Ablation study on each component. After adversarial fine-tuning the model using adversarial examples generated by PGD-2, we verify the robustness of the model using adversarial examples generated by PGD-100.

|  | Robust | Clean | Average |
|---|---|---|---|
| CLIP | 4.90 | 64.42 | 34.66 |
| $L_{CE}$ | 29.45 | 44.97 | 37.21 |
| $+L_{AR}$ | 31.71 | 49.96 | 40.84 |
| $+L_{AMC}$ | 41.96 | 56.48 | 49.22 |

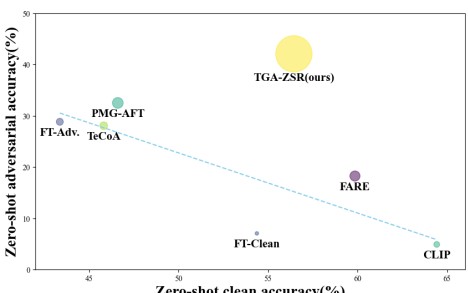

Figure 3: The trade-off between robustness and clean accuracy. Each point on the graph represents a method, with the size of the point indicating the extent to which it achieves a favorable trade-off between robustness and clean accuracy.

Table 8: Comparison of memory usage, training time, and test time.

| Methods | Train memory usage | Train time (per epoch / batch) | Test time (per batch) |
|---|---|---|---|
| CLIP [48] | 0Mb | 0s / 0s | 21s |
| TeCoA [38] | 12873Mb | 512s / 0.65s | 21s |
| PMG-AFT[59] | 18449Mb | 828s / 1.06s | 21s |
| TGA-ZSR (ours) | 21227Mb | 885s / 1.13s | 21s |

mately 15% compared to state-of-the-art method PMG-AFT. This is due to the additional computation required for the text-guided attention map. The training time for our method is comparable to that of PMG-AFT. The test time remains consistent across all methods.

# 5 Conclusion and Limitations

In this paper, we discovered that adversarial attacks lead shift of text-guided attention. Building on this observation, we introduce a text-guided approach, TGA-ZSR, which incorporates two key components to preform adversarial fine-tuning and constrain the model. This strategy prevents model drift while enhancing model robustness. Extensive experiments validate the performance of TGA-ZSR, which not only improves CLIP's zero-shot adversarial robustness but also maintains zero-shot clean accuracy on clean examples, gaining a favorable balance.

**Limitations.** We use a simple text-guided attention mechanism by multiplying the text embedding and vision embedding which is effective against most attack types. However, for more challenging attacks such as AutoAttack, the improvement remains limited. This indicates that while our approach shows promise, it may require further refinement to enhance robustness under stronger adversarial scenarios.

**Border Impact.** Large-scale pre-trained vision-language models (VLMs) like CLIP [48] integrate visual and textual data, revolutionizing applications such as image classification, semantic segmentation, and vision question answering. While these models excel in zero-shot learning and transfer learning, they are vulnerable to adversarial attacks, posing risks in critical applications like autonomous vehicles and medical diagnosis. Adversarial training improves robustness but has practical challenges, including increased computational overhead and potential overfitting. Exploring zero-shot adversarial robustness is essential to ensure reliability.

**Acknowledgement.** This work was supported by National Science and Technology Major Project under Grant 2021ZD0112200, in part by the National Natural Science Foundation of China under Grants 62202331, U23A20387, 62036012, 62276118.

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

# A  Appendix

## A.1  Experiments on More Datasets.

**Experiments on ImageNet_subset.**  We follow the state-of-the-art method PMG-AFT, which was fine-tuned on Tiny-ImageNet. In addition to this, we further evaluate our method on the ImageNet_subset (a random selection of 100 classes from the full ImageNet dataset). The results are shown in Table 9 and Table 10. For adversarial robustness, our approach achieves optimal results on several datasets and sub-optimal results on the remaining datasets, with an overall performance that is approximately 1% higher than the previous state-of-the-art. In terms of clean accuracy, our method performs worse than the generalization-focused FARE but outperforms other methods.

Table 9: Zero-shot robust accuracy on images attacked with 100 steps of PGD [36]. We performed several different methods on ImageNet_subset and evaluated across 16 datasets. The optimal accuracy is highlighted in **bold**, while the second-best accuracy is underlined.

| Methods | Tiny-ImageNet | CIFAR-10 | CIFAR-100 | STL-10 | SUN397 | Food101 | Oxfordpets | Flowers102 | DTD | EuroSAT | FGVC-Aircraft | ImageNet | Caltech-101 | Caltech-256 | StanfordCars | PCAM | Average |
|---|---|---|---|---|---|---|---|---|---|---|---|---|---|---|---|---|---|
| CLIP [48] | 0.88 | 0.42 | 0.26 | 26.11 | 1.00 | 6.60 | 3.84 | 1.19 | 2.02 | 0.05 | 0.00 | 1.24 | 19.88 | 12.60 | 0.20 | 0.11 | 4.90 |
| TeCoA [38] | 7.83 | 19.23 | 7.30 | 51.76 | 16.38 | 10.06 | 32.03 | 25.00 | 13.67 | 11.13 | 3.51 | 17.92 | 45.19 | 33.45 | 8.76 | 23.59 | 20.42 |
| FARE[51] | 3.66 | 6.76 | 3.33 | 45.16 | 4.99 | 8.13 | 13.90 | 8.57 | 9.63 | 4.82 | 0.36 | 7.82 | 33.16 | 25.86 | 1.79 | 4.65 | 11.41 |
| PMG-AFT[59] | **12.16** | 24.75 | **12.06** | 57.13 | **20.68** | **15.13** | 40.47 | **29.62** | 13.67 | 11.40 | **3.84** | **21.62** | **51.48** | **39.24** | **11.70** | 17.88 | 23.93 |
| TGA-ZSR(ours) | 10.13 | **28.37** | 10.92 | **61.37** | 19.61 | 13.63 | **40.61** | 25.39 | **14.63** | **11.58** | 3.42 | 20.28 | 49.48 | 38.95 | 10.40 | **37.09** | **24.74** |

Table 10: Zero-shot clean accuracy. We performed several different methods on ImageNet_subset and evaluated across 16 datasets.

| Methods | Tiny-ImageNet | CIFAR-10 | CIFAR-100 | STL-10 | SUN397 | Food101 | Oxfordpets | Flowers102 | DTD | EuroSAT | FGVC-Aircraft | ImageNet | Caltech-101 | Caltech-256 | StanfordCars | PCAM | Average |
|---|---|---|---|---|---|---|---|---|---|---|---|---|---|---|---|---|---|
| CLIP [48] | **57.26** | **88.06** | **60.45** | **97.04** | **57.26** | **83.89** | **87.41** | **65.47** | **40.69** | **42.59** | **20.25** | **59.15** | **85.34** | **81.73** | **52.02** | 52.09 | **64.42** |
| TeCoA [38] | 22.21 | 48.78 | 21.00 | 78.41 | 40.86 | 29.56 | 67.95 | 44.15 | 24.26 | 11.69 | 10.74 | 38.19 | 69.93 | 58.15 | 31.00 | 54.03 | 40.68 |
| FARE[51] | 55.68 | 85.11 | 58.47 | 93.11 | 53.64 | 76.99 | 83.88 | 62.27 | 35.18 | 31.33 | 15.91 | 56.18 | 82.26 | 78.27 | 46.84 | 44.92 | 60.00 |
| PMG-AFT[59] | 28.02 | 53.66 | 26.95 | 80.58 | 43.70 | 36.96 | 70.51 | 47.61 | 22.66 | 13.65 | 10.23 | 40.66 | 74.17 | 61.27 | 31.70 | 47.27 | 43.10 |
| TGA-ZSR(ours) | 29.16 | 67.46 | 30.81 | 86.30 | 47.65 | 41.30 | 73.56 | 47.02 | 25.96 | 15.33 | 11.46 | 43.13 | 75.48 | 65.19 | 35.27 | **55.31** | 46.90 |

**Results on Diverse Datasets.**  In addition to validate the method on 16 datasets following the evaluation protocol of previous works, we also conduct evaluations on three additional datasets: ImageNet-A (natural adversarial examples) [20], ImageNet-O (out-of-distribution data) [20], and ImageNet-R (multiform art form) [19]. In Table 11 and Table 12, the clean accuracy on ImageNet-A is notably lower compared to ImageNet-O and ImageNet-R . In general, other methods enhance robust accuracy by around 2%, they also incur a steep 20% drop in clean accuracy. In contrast, our approach achieves a notable 5% improvement in zero-shot robust accuracy, although with a 16% decrease in clean accuracy, outperforming other methods. Although our method doesn't excel in zero-shot clean accuracy for ImageNet-O and ImageNet-R, it excels in optimizing zero-shot robust accuracy. Overall, our method yields the best results across these three datasets, as demonstrated by the *Average*.

Table 11: Zero-shot robust accuracy on images attacked with 100 steps of PGD [36]. We performed different several methods on Tiny-ImageNet and evaluated in the following three additional datasets. The optimal accuracy is highlighted in **bold**, while the second-best accuracy is underlined.

| Methods | ImageNet-A | ImageNet-O | ImageNet-R | Average |
|---|---|---|---|---|
| CLIP [48] | 0.08 | 0.60 | 5.57 | 2.08 |
| FT-Clean | 0.17 | 1.55 | 4.67 | 2.13 |
| FT-Adv. | 1.91 | 22.30 | 21.92 | 15.37 |
| TeCoA [38] | 1.67 | 23.55 | 23.85 | 16.36 |
| FARE [51] | 0.72 | 9.00 | 18.99 | 9.57 |
| PMG-AFT [59] | 2.35 | 27.70 | 27.61 | 19.22 |
| TGA-ZSR (ours) | **5.03** | **32.10** | **36.87** | **24.67** |

Table 12: Zero-shot clean accuracy. We performed different several methods on Tiny-ImageNet and evaluated in the following three additional datasets. The optimal accuracy is highlighted in **bold**, while the second-best accuracy is underlined.

| Methods | ImageNet-A | ImageNet-O | ImageNet-R | Average |
|---|---|---|---|---|
| CLIP [48] | **29.49** | 46.05 | **63.61** | **46.38** |
| FT-Clean | 17.23 | 39.25 | 50.68 | 35.72 |
| FT-Adv. | 6.45 | 37.25 | 35.80 | 26.50 |
| TeCoA [38] | 6.04 | 42.50 | 40.65 | 29.73 |
| FARE [51] | 16.24 | **49.30** | 58.62 | 41.39 |
| PMG-AFT [59] | 6.19 | 42.55 | 40.81 | 29.85 |
| TGA-ZSR (ours) | 13.77 | 47.30 | 52.39 | 37.82 |

## A.2 More Ablation Studies

**Trade-off between $\alpha$ and $\beta$.** Following the protocol of previous works (TeCoA [38], PMG-AFT [59], FARE [51]), we fine-tuned the CLIP model on adversarial samples from a single dataset (Tiny-ImageNet in our case) for 'adversarial fine-tuning' and subsequently evaluated its performance across 15 datasets, including Tiny-ImageNet itself. Thus we only need to tune hyperparameters on just the training dataset. We randomly selected 80% of the training set for training and the remaining 20% for validation to choose the hyperparameters. The validation set results are shown in Table 13. The final results on the test set were obtained by training on the entire training set using the optimal hyperparameters ($\alpha$=0.08, $\beta$=0.05) identified from the validation set.

Table 13: Results on validation set of Tiny-ImageNet dataset.

| Hyper-parameters | Robust | Clean | Average |
|---|---|---|---|
| $\alpha = 0.07, \beta = 0.05$ | 64.32 | 75.92 | 70.12 |
| $\alpha = 0.08, \beta = 0.04$ | 47.25 | 76.20 | 61.72 |
| $\alpha = 0.08, \beta = 0.06$ | 58.28 | 76.08 | 67.18 |
| $\alpha = 0.09, \beta = 0.05$ | 46.20 | 76.10 | 61.15 |
| $\alpha = 0.08, \beta = 0.05$ | 64.01 | 77.79 | 70.90 |

After choosing the optimal hyper-parameter on the validation set of the Tiny-ImageNet dataset, we proceeded to analyze the sensitivity of these hyper-parameters on the overall performance across 16 datasets, including the Tiny-ImageNet and 15 additional datasets. This step was crucial for evaluating the robustness and generalizability of the model under different hyper-parameter settings. Table 14 and Table 15 demonstrate the results of different hyper-parameter of $\alpha$ and $\beta$ in Eq. 8.

Table 14: Zero-shot robust accuracy on images attacked with 100 steps of PGD [36]. We compared the results of different hyper-parameters on Tiny-ImageNet and evaluated across 16 datasets. The optimal accuracy is highlighted in **bold**, while the second-best accuracy is underlined.

| Hyper-parameters | Tiny-ImageNet | CIFAR-10 | CIFAR-100 | STL-10 | SUN397 | Food101 | Oxfordpets | Flowers102 | DTD | EuroSAT | FGVC-Aircraft | ImageNet | Caltech-101 | Caltech-256 | StanfordCars | PCAM | Average |
|---|---|---|---|---|---|---|---|---|---|---|---|---|---|---|---|---|---|
| $\alpha$=0.07, $\beta$=0.05 | 62.27 | 57.87 | 33.48 | 82.61 | 30.95 | 31.89 | 54.87 | 33.23 | **22.50** | 13.37 | **4.56** | 27.35 | 68.70 | 58.05 | 18.52 | 46.63 | 40.43 |
| $\alpha$=0.09, $\beta$=0.05 | 48.43 | 40.68 | 22.51 | 73.75 | 23.16 | 19.34 | 42.27 | 25.87 | 18.25 | 12.47 | 3.12 | 21.39 | 59.83 | 48.87 | 12.29 | 44.71 | 32.31 |
| $\alpha$=0.08, $\beta$=0.03 | 49.62 | 40.72 | 22.81 | 72.09 | 21.69 | 16.75 | 40.72 | 23.39 | 17.18 | 11.64 | 3.18 | 20.39 | 58.33 | 46.83 | 11.40 | **48.54** | 31.58 |
| $\alpha$=0.08, $\beta$=0.04 | 57.40 | 53.00 | 30.49 | 79.44 | 26.75 | 26.07 | 49.58 | 30.09 | 20.90 | 13.26 | 3.93 | 24.17 | 66.61 | 54.96 | 15.14 | 44.67 | 37.28 |
| $\alpha$=0.08, $\beta$=0.06 | 61.01 | 57.29 | 32.29 | 81.94 | 31.12 | 31.52 | 55.17 | 33.00 | 21.60 | 13.84 | 4.41 | 27.17 | 69.39 | 58.05 | 17.55 | 47.22 | 40.16 |
| $\alpha$=0.08, $\beta$=0.05 | **63.97** | **61.82** | **35.25** | **83.99** | 32.78 | **34.13** | 56.91 | **34.20** | 21.92 | **14.20** | 4.44 | **28.62** | **70.53** | **59.70** | 21.15 | 47.75 | **41.96** |

Table 15: Zero-shot clean accuracy. We compared the results of different hyper-parameters on Tiny-ImageNet and evaluated across 16 datasets. The optimal accuracy is highlighted in **bold**, while the second-best accuracy is underlined.

| Hyper-parameters | Tiny-ImageNet | CIFAR-10 | CIFAR-100 | STL-10 | SUN397 | Food101 | Oxfordpets | Flowers102 | DTD | EuroSAT | FGVC-Aircraft | ImageNet | Caltech-101 | Caltech-256 | StanfordCars | PCAM | Average |
|---|---|---|---|---|---|---|---|---|---|---|---|---|---|---|---|---|---|
| $\alpha$=0.07, $\beta$=0.05 | 76.91 | 86.64 | **57.14** | **93.39** | 51.87 | 58.84 | 77.46 | 48.97 | 29.89 | 25.24 | 11.82 | 48.45 | 80.85 | 75.14 | 37.46 | 49.67 | 56.86 |
| $\alpha$=0.09, $\beta$=0.05 | 76.05 | 85.04 | 55.46 | 92.48 | 47.36 | 52.50 | 74.63 | 45.37 | **30.59** | 24.69 | 10.83 | 44.98 | 77.85 | 72.35 | 34.01 | **49.98** | 54.63 |
| $\alpha$=0.08, $\beta$=0.03 | 76.97 | 80.71 | 49.49 | 91.33 | 45.02 | 42.62 | 70.16 | 41.15 | 26.92 | 18.97 | 9.93 | 41.16 | 77.26 | 68.24 | 30.36 | 49.91 | 51.26 |
| $\alpha$=0.08, $\beta$=0.04 | **77.20** | 85.60 | 56.03 | 93.16 | 51.06 | 56.55 | 77.19 | 46.04 | 29.47 | 23.97 | 11.64 | 47.33 | **81.37** | 74.14 | 35.23 | 49.34 | 55.96 |
| $\alpha$=0.08, $\beta$=0.06 | 76.67 | **86.75** | 56.80 | 93.33 | 51.77 | 58.76 | **77.62** | **49.10** | 29.79 | **26.24** | 11.43 | 48.63 | 81.09 | 75.28 | 37.11 | 49.64 | 56.88 |
| $\alpha$=0.08, $\beta$=0.05 | 76.85 | 86.23 | 56.55 | 93.28 | 51.71 | 57.72 | 77.08 | 48.32 | 29.15 | 23.99 | **12.03** | 48.10 | 80.82 | 74.58 | **37.72** | 49.60 | 56.48 |

**Effect of Each Component.** Due to space constraints, a detailed experiment on the effect of each component was not provided in Table 7. Therefore, we present here a comprehensive experiment detailing the effect of each component. To explore the impact of each component on the final model performance, we conducted a series of ablation experiments to evaluate the effectiveness of each component. From the experimental results Table 16 and Table 17, it's clear that the inclusion of our text-guided attention components enhances CLIP's zero-shot robust accuracy and maintains its zero-shot clean accuracy.

Table 16: Zero-shot robust accuracy on images attacked with 100 steps of PGD. After adversarial fine-tuning the model using adversarial examples generated by PGD-2, we performed each component and evaluated across 16 datasets. The optimal accuracy is highlighted in **bold**.

| Components | Tiny-ImageNet | CIFAR-10 | CIFAR-100 | STL-10 | SUN397 | Food101 | Oxfordpets | Flowers102 | DTD | EuroSAT | FGVC-Aircraft | ImageNet | Caltech-101 | Caltech-256 | StanfordCars | PCAM | Average |
|---|---|---|---|---|---|---|---|---|---|---|---|---|---|---|---|---|---|
| $L_{CE}$ | 52.88 | 40.36 | 22.59 | 70.23 | 19.53 | 13.66 | 37.64 | 19.79 | 16.76 | 11.79 | 2.64 | 18.11 | 55.55 | 42.45 | 8.89 | 38.32 | 29.45 |
| $+L_{AR}$ | 51.08 | 41.76 | 23.39 | 72.49 | 22.01 | 16.44 | 40.12 | 22.10 | 18.09 | 11.62 | 3.06 | 20.24 | 58.70 | 46.56 | 11.47 | **48.20** | 31.71 |
| $+L_{AMC}$ | **63.97** | **61.82** | **35.25** | **83.99** | **32.78** | **34.13** | **56.91** | **34.20** | **21.92** | **14.20** | **4.44** | **28.62** | **70.53** | **59.70** | **21.15** | 47.75 | **41.96** |

Table 17: Zero-shot clean accuracy. After adversarial fine-tuning the model using adversarial examples generated by PGD-2, we verify the robustness of the model using adversarial examples generated by PGD-100. The optimal accuracy is highlighted in **bold**.

| Components | Tiny-ImageNet | CIFAR-10 | CIFAR-100 | STL-10 | SUN397 | Food101 | Oxfordpets | Flowers102 | DTD | EuroSAT | FGVC-Aircraft | ImageNet | Caltech-101 | Caltech-256 | StanfordCars | PCAM | Average |
|---|---|---|---|---|---|---|---|---|---|---|---|---|---|---|---|---|---|
| $L_{CE}$ | 74.23 | 70.06 | 41.05 | 87.40 | 36.90 | 30.02 | 62.55 | 33.76 | 23.88 | 16.20 | 5.85 | 33.58 | 72.27 | 59.69 | 20.59 | **50.92** | 44.97 |
| $+L_{AR}$ | 76.71 | 77.91 | 47.67 | 90.21 | 44.17 | 39.46 | 69.26 | 39.29 | 26.28 | 18.14 | 8.04 | 39.55 | 76.76 | 66.98 | 29.00 | 49.95 | 49.96 |
| $+L_{AMC}$ | **76.85** | **96.23** | **56.55** | **93.28** | **51.71** | **57.72** | **77.08** | **48.32** | **29.15** | **23.99** | **12.03** | **48.10** | **80.82** | **74.58** | **37.72** | 49.60 | **56.48** |

**Effect of Distance Metrics on Loss Function.** Except $l_2$, $cosine$ and $l_1$ are also frequently utilized distance metrics. We compared the performance of our method using these distance metrics. The results from Table 18 and Table 19 demonstrate that $cosine$ and $l_1$ exhibit similar performance but are inferior to $l_2$, except for the zero-shot clear accuracy on PCAM. $l_2$ outperforms the other two distance measures by enhancing zero-shot adversarial robustness and zero-shot clean accuracy by approximately 12% and 11%, respectively. Thus we choose $l_2$ distance to measure in our loss function.

**Effect of Learning Rate.** Learning rate stands as a significant hyper-parameter in model training. Here we validate the effect of the learning rate for the experiment in Table 20 and Table 21. When the learning rate is set to 0.001, we observe the lowest values for both zero-shot robust accuracy and zero-shot clean accuracy. And, when we reduce the learning rate to 0.00001, we note that CLIP's zero-shot clean accuracy remains relatively stable, demonstrating the highest performance in this experiment. However, it affects CLIP's zero-shot robust accuracy, resulting in a less favorable balance. Selecting a learning rate of 0.0001 achieved a well-balanced improvement, with the average of zero-shot robust accuracy and zero-shot clean accuracy increasing by 18.48% and 6.41%, respectively, compared to the learning rates of 0.001 and 0.00001.

Table 18: Zero-shot robust accuracy on images attacked with 100 steps of PGD [36]. We compared the results of different distance metrics on Tiny-ImageNet and evaluated across 16 datasets. The optimal accuracy is highlighted in **bold**.

| Distance metrics | Tiny-ImageNet | CIFAR-10 | CIFAR-100 | STL-10 | SUN397 | Food101 | Oxfordpets | Flowers102 | DTD | EuroSAT | FGVC-Aircraft | ImageNet | Caltech-101 | Caltech-256 | StanfordCars | PCAM | Average |
|---|---|---|---|---|---|---|---|---|---|---|---|---|---|---|---|---|---|
| $cos$ | 52.87 | 40.34 | 22.82 | 70.40 | 19.55 | 13.63 | 37.59 | 19.63 | 16.81 | 11.77 | 2.70 | 18.11 | 55.72 | 42.45 | 8.83 | 39.02 | 29.57 |
| $l_1$ | 52.94 | 40.46 | 22.82 | 70.45 | 19.57 | 13.69 | 37.69 | 19.74 | 17.18 | 11.75 | 2.76 | 18.09 | 55.66 | 42.47 | 8.93 | 38.46 | 29.54 |
| Ours ($l_2$) | **63.97** | **61.82** | **35.25** | **83.99** | **32.78** | **34.13** | **56.91** | **34.20** | **21.92** | **14.20** | **4.44** | **28.62** | **70.53** | **59.70** | **21.15** | **47.75** | **41.96** |

Table 19: Zero-shot clean accuracy. We compared the results of different distance metrics on Tiny-ImageNet and evaluated across 16 datasets. The optimal accuracy is highlighted in **bold**.

| Distance metrics | Tiny-ImageNet | CIFAR-10 | CIFAR-100 | STL-10 | SUN397 | Food101 | Oxfordpets | Flowers102 | DTD | EuroSAT | FGVC-Aircraft | ImageNet | Caltech-101 | Caltech-256 | StanfordCars | PCAM | Average |
|---|---|---|---|---|---|---|---|---|---|---|---|---|---|---|---|---|---|
| $cos$ | 74.33 | 70.84 | 41.16 | 87.31 | 37.02 | 30.08 | 62.72 | 34.04 | 24.10 | 16.24 | 5.88 | 33.61 | 72.40 | 59.72 | 20.51 | 50.64 | 45.04 |
| $l_1$ | 74.38 | 71.03 | 41.32 | 87.38 | 37.06 | 30.12 | 62.55 | 33.99 | 23.83 | 16.23 | 5.76 | 33.60 | 72.33 | 59.71 | 20.68 | **50.88** | 45.05 |
| Ours ($l_2$) | **76.85** | **86.23** | **56.55** | **93.28** | **51.71** | **57.72** | **77.08** | **48.32** | **29.15** | **23.99** | **12.03** | **48.10** | **80.82** | **74.58** | **37.72** | 49.60 | **56.48** |

Table 20: Zero-shot robust accuracy on images attacked with 100 steps of PGD [36]. We compared the results of different learning rates on Tiny-ImageNet and evaluated across 16 datasets. The optimal accuracy is highlighted in **bold**.

| Learning Rate | Tiny-ImageNet | CIFAR-10 | CIFAR-100 | STL-10 | SUN397 | Food101 | Oxfordpets | Flowers102 | DTD | EuroSAT | FGVC-Aircraft | ImageNet | Caltech-101 | Caltech-256 | StanfordCars | PCAM | Average |
|---|---|---|---|---|---|---|---|---|---|---|---|---|---|---|---|---|---|
| 0.001 | 52.19 | 37.27 | 18.10 | 61.36 | 12.21 | 8.42 | 26.74 | 10.41 | 11.28 | 2.26 | 1.26 | 12.96 | 40.27 | 29.64 | 2.95 | 39.72 | 22.94 |
| 0.00001 | 26.16 | 20.67 | 11.80 | 63.00 | 15.25 | 13.34 | 30.55 | 22.57 | 14.47 | 10.84 | 2.10 | 15.05 | 52.60 | 40.09 | 6.65 | 44.62 | 24.36 |
| 0.0001 | **63.97** | **61.82** | **35.25** | **83.99** | **32.78** | **34.13** | **56.91** | **34.20** | **21.92** | **14.20** | **4.44** | **28.62** | **70.53** | **59.70** | **21.15** | **47.75** | **41.96** |

Table 21: Zero-shot clean accuracy. We compared the results of different learning rates on Tiny-ImageNet and evaluated across 16 datasets. The optimal accuracy is highlighted in **bold**.

| Learning Rate | Tiny-ImageNet | CIFAR-10 | CIFAR-100 | STL-10 | SUN397 | Food101 | Oxfordpets | Flowers102 | DTD | EuroSAT | FGVC-Aircraft | ImageNet | Caltech-101 | Caltech-256 | StanfordCars | PCAM | Average |
|---|---|---|---|---|---|---|---|---|---|---|---|---|---|---|---|---|---|
| 0.001 | 75.32 | 69.83 | 37.96 | 82.45 | 28.17 | 21.68 | 50.45 | 22.93 | 19.10 | 13.04 | 4.23 | 25.48 | 59.84 | 48.18 | 8.73 | 49.18 | 38.54 |
| 0.00001 | 68.58 | **87.49** | **60.14** | **94.68** | **56.97** | **72.92** | **81.55** | **58.55** | **33.94** | **35.09** | **16.26** | **54.89** | **83.15** | **80.07** | **46.60** | 49.32 | **61.26** |
| 0.0001 | **76.85** | 86.23 | 56.55 | 93.28 | 51.71 | 57.72 | 77.08 | 48.32 | 29.15 | 23.99 | 12.03 | 48.10 | 80.82 | 74.58 | 37.72 | **49.60** | 56.48 |

