# OpenReview forum: "Text-Guided Attention is All You Need for Zero-Shot Robustness in Vision-Language Models"
_NeurIPS.cc/2024/Conference — NeurIPS 2024 poster_

### Official Review · Reviewer_mEBo · 2024-07-01

**Soundness:** 2
**Presentation:** 4
**Contribution:** 2
**Rating:** 5
**Confidence:** 4

**Summary:**

The paper focuses on enhancing the robustness of CLIP models against adversarial perturbations. The approach utilizes saliency maps generated by the inner products between image and text embeddings. Two regularization terms are introduced: the first aims to align the saliency maps of the original and its adversarial examples, while the second seeks to match the saliency maps of the original example between the pre-trained CLIP model and the target model. Experimental results indicate that fine-tuning CLIP with these two regularization terms improves its robustness to attacks.

**Strengths:**

1.	The paper is well-written and organized, with clear descriptions of the motivation and methods.
2.	To the best of my knowledge, enhancing CLIP robustness by aligning the saliency maps produced through inner products between image and text embeddings, termed "Text-Guided Attention" in the paper, is a novel approach.
3.	It is good to see the detailed ablation analysis.

**Weaknesses:**

1.	The abstract claims that the goal is to enhance both the generalization and robustness of the CLIP model; however, the experimental results do not fully support this assertion. Specifically, regarding generalization, Table 2 shows a significant decrease in clean accuracy compared to the original pre-trained CLIP model. I kindly ask the authors to use the terms "generalizability" and "adversarial robustness" more carefully within the paper. Typically, the generalizability of CLIP models refers to their zero-shot classification performance across different datasets. It may be argued that the proposed method improves the adversarial robustness of CLIP models while maintaining relatively good generalizability compared to previous adversarial training methods. However, the authors must acknowledge the decrease in generalizability compared to original CLIP model as clearly evidenced by the results in Table 2.

2. Since the training uses adversarial examples produced by PGD, it is not surprising that the proposed method results in a model with good robustness to PGD attacks, as shown in Table 1. However, when the evaluation comes to AutoAttack, the advantage of the TGA-ZSR over the previously best method is marginal as shown in Table 3. Could the authors provide explanations for this point, and could they demonstrate the robustness of the resulting models to more attacks?

**Questions:**

See weaknesses.

**Limitations:**

The primary limitation of the paper is that the experimental results presented do not sufficiently support its claims. I recommend that the authors either revise the claims to align more closely with the data presented or extend the experimental section to provide additional evidence supporting their assertions.

---

> ### Author Rebuttal · Authors · 2024-08-06
>
> **Q1: Use the terms "generalizability" and "adversarial robustness" more carefully.**
>
> **A1**: Thank you for pointing that out! The reviewer is correct. To balance the performance between clean and adversarial samples, our goal is to maintain the generalization and enhance the adversarial robustness of the original CLIP model. We will modify it in the revised version of the paper.
>
> **Q2: The advantage of the proposed method is less on AutoAttack than PGD.**
>
> **A2**: In the paper, we train our model using the PGD attack and validate our approach on both PGD and AutoAttack. Thus, testing on AutoAttack serves as a cross-validation experiment. Additionally, AutoAttack combines multiple attack strategies, effectively covering a wide range of defense techniques and model architectures. It can adapt to the characteristics and defense strategies of the target model, making its attacks more powerful. For these reasons, AutoAttack generally results in lower adversarial accuracy compared to methods like PGD, and improving model performance against AutoAttack can be challenging. Consequently, our method shows less improvement on AutoAttack, but we still achieved state-of-the-art performance.
>
> **Q3: Results on more attacks.**
>
> **A3**: To further evaluate the effectiveness of the proposed method, as suggested by the reviewer, we conducted experiments using another type of attack, specifically the CW attack [c]. The results, shown below, demonstrate that our method significantly outperforms the state-of-the-art method PMG-AFT on both adversarial and clean samples. This substantial margin indicates the adversarial robustness of our proposed method.
>
> Table 7: Zero-shot adversarial robust accuracy and clean accuracy across 16 datasets  with **CW attack**.
>
> |  Test  |Methods | Tiny- |  C.10  |  C.100 |   STL  |   SUN   |   Food  | Pets | Flowers |     DTD    |   EuroS.  | Airc.|  ImageN.  | Ca.101 | Ca.256 | Cars |    PCAM    |   Avg.  |
> |:------:|:-------:|:-------------:|:--------:|:---------:|:------:|:------:|:-------:|:----------:|:----------:|:-----:|:-------:|:-------------:|:--------:|:-----------:|:-----------:|:------------:|:-----:|:-------:|
> |        |   CLIP  |      0.21     |   0.36   |    0.10   |  10.59 |  1.16  |   0.82  |    1.23    |    1.09    |  2.18 |   0.01  |      0.00     |   1.14   |    13.50    |     7.36    |     2.36     |  0.07 |   3.64  |
> | Robust | PMG-AFT |     44.59     |   44.86  |   24.15   |  74.11 |  19.99 |  17.33  |    39.88   |    20.95   | 13.51 |  12.09  |      1.47     |   19.51  |    60.99    |    44.46    |     10.57    | 48.59 |  31.07  |
> |        |   Ours  |     63.85     |   60.50  |   34.62   |  84.11 |  22.03 |  33.28  |    58.33   |    32.95   | 21.22 |  13.89  |      4.56     |   20.42  |    70.34    |    59.73    |     20.20    | 48.02 |  40.50  |
> |
> |        |   CLIP  |     57.26     |   88.06  |   60.45   |  97.04 |  57.26 |  83.89  |    87.41   |    65.47   | 40.69 |  42.59  |     20.25     |   59.15  |    85.34    |    81.73    |     52.02    | 52.09 |  64.42  |
> |  Clean | PMG-AFT |     66.98     |   74.50  |   44.66   |  88.95 |  37.31 |  37.42  |    66.12   |    35.65   | 21.49 |  18.02  |      4.53     |   35.92  |    76.82    |    61.95    |     25.00    | 49.97 |  46.58  |
> |        |   Ours  |     76.99     |   86.24  |   56.11   |  93.66 |  47.12 |  56.80  |    77.65   |    47.10   | 28.94 |  24.64  |     11.52     |   43.89  |    80.65    |    74.76    |     35.67    | 49.78 | 55.72  |
>
> [c] Carlini, Nicholas and David A. Wagner. “Towards Evaluating the Robustness of Neural Networks.” 2017 IEEE Symposium on Security and Privacy (SP) (2016): 39-57.
>
> **We will revise and add the corresponding context in the final version.**

---

> > ### Comment · Reviewer_mEBo · 2024-08-08
> >
> > The author have addressed my concern, and I will change my score from 4 to 5.

---

> > > ### Author Response · Authors · 2024-08-09
> > >
> > > Thank you for taking the time to read our response and increasing your score! We are glad to hear that the response addressed your concerns.

---

### Official Review · Reviewer_BFe6 · 2024-07-06

**Soundness:** 3
**Presentation:** 4
**Contribution:** 3
**Rating:** 6
**Confidence:** 5

**Summary:**

This work studies the robustness to adversarial samples for CLIP. Inspired by an observation that adversarial perturbations induce shifts in text-guided attention, the work proposes a simple yet effective approach to improve the zero-shot robostness, i.e., align the text-guided attention of clean samples and adversarial samples. The state-of-the-art performance across 16 datasets demonstrate the effectiveness of the proposed method.

**Strengths:**

This work focuses on an interesting topic, which involves the robustness of VLMs. To improve the zero-shot robustness of CLIP, the work proposes a text-guided attention-based method, which is simple but intuitive and effective. Extensive experiments on 16 datasets demonstrate the effectiveness of the proposed method. Additionally, the paper is well written and the experiments are solid.

**Weaknesses:**

I would like to see more comparison with other types of attention, e.g., vision-only self-attention. Does the proposed method works because of the attention mechanism or the text guidance?

**Questions:**

1. How do you utilize the class token in the ViT backbone of CLIP?
2. In Tables 13 and 14, why the losses L1 and L2 have very different performance?

**Limitations:**

The paper has discussed the limitations and potential impacts of the work.

---

> ### Author Rebuttal · Authors · 2024-08-06
>
> **Q1: More comparison with other types of attention.**
>
> **A1**: We also considered this point when exploring the comprehensive effect of attention. Thus, we conducted an experiment by replacing the text-guided attention with vision-only attention, as detailed in the first part of Section 4.4 ("Different types of attention"). We used Grad-CAM to generate the vision-only attention map. The results in Table 4 demonstrate that vision-based attention already achieves results comparable to the state-of-the-art method PMG-AFT in terms of both zero-shot adversarial robustness accuracy and clean accuracy, although lower than those achieved with text-guided attention. This indicates that the zero-shot adversarial robustness of vision-language models benefits from the constraints of attention mechanisms, with text-guided attention further enhancing performance. We will explore more types of attention in future analysis.
>
> **Q2: How to use class token?**
>
> **A2**: In Figure 2, the class token used as  $f(x_a)$ is obtained from $f_g^{tar}(x_a)$ after pooling. We will clarify this in more detail in future versions.
>
> **Q3: In Tables 13 and 14, why the losses L1 and L2 have very different performance.**
>
> **A3**: We investigated the use of L1 and L2 norms to constrain the text-guided attention map. The results indicated that the L2 norm significantly outperforms the L1 norm. The primary reason for this is that the L2 norm, tends to distribute the penalty more evenly across all elements of the attention map. This leads to smoother and more stable attention distributions, which can enhance the model's adversarial robustness.
>
> In contrast, the L1 norm, tends to produce sparser solutions by penalizing non-zero elements more heavily. While this can be beneficial in certain contexts by promoting sparsity, it may lead to less stable and less coherent attention maps in the case of text-guided attention, ultimately degrading performance.
>
> **We will revise and add the corresponding context in the final version.**

---

> > ### Comment · Reviewer_BFe6 · 2024-08-11
> >
> > Thanks for the authors' response. My concerns have been addressed.
> >
> > Additionally, I have one more question about the proposed model: Why introducing the text guidance can significantly improve the robustness? I think this work is insightful and the proposed model is interesting, and thus I would like to see more analysis or more insights.
> >
> > If the authors can address this, I am willing to increasing my rating.

---

> ### Author Response · Authors · 2024-08-11
>
> Thank you for taking the time to review our response and for recognizing the insights and interest in our work. We are pleased to hear that our previous response addressed your concerns.
>
> Regarding your new question, we believe that the significant improvement in robustness with text guidance can be attributed to two key aspects:
>
> (1) **Selective Focus on Relevant Features**: Text-guided attention allows the model to selectively focus on the most relevant parts of the image based on the text input. For instance, if the text describes "a photo of an apple," the attention mechanism guides the model to focus on the apple in the image, while ignoring irrelevant details. This selective focus enables the model to extract the most pertinent information, thus enhancing robustness by minimizing distractions from non-essential features.
>
> To illustrate the advantage of text-guided attention over vision-only attention, we calculated the mean Intersection over Union (mIoU) of the attention maps generated by both methods on the validation set of the ImageNet-S-50 dataset [d], which includes ground truth masks for semantic segmentation tasks. The results below demonstrate that the mIoU of text-guided attention is significantly higher than that of vision-only attention, indicating the superior effectiveness of our approach.
>
> Table 8: mIoU Comparison of Text-Guided and Vision-Only Attention on ImageNet-S-50 Dataset.
> | Method | mIoU |
> |----------| -------|
> |     Text-guided    | **0.485** |
> |   Vision-only  | 0.400 |
>
> (2) **Consistency Across Modalities**: By integrating text-guided attention, the model maintains consistency between the visual and textual modalities. The attention mechanism aligns visual features with corresponding textual descriptions, reinforcing the semantic connections between the two. This alignment reduces the risk of the model making inconsistent or erroneous predictions based on a single modality, thereby enhancing overall robustness.
>
> To further support this point, we conducted an experiment measuring the difference (L2 distance) in attention maps before and after an adversarial attack using both the text-guided attention-based model and the vision-only model on the TinyImageNet validation set. The results below show that the text-guided attention model maintains better consistency after the attack compared to the vision-only model.
>
> Table 9: L2 Distance Comparison Before and After Attack on TinyImageNet.
> | Method | L2 distance|
> |----------| -------|
> |     Text-guided    | **29.886** |
> |   Vision-only  | 52.918 |
>
> Based on these findings, we believe that the introduction of text guidance significantly enhances the robustness of the model.
>
> We hope this response addresses your concerns.
>
> [d] Shanghua Gao, Zhong-Yu Li, Ming-Hsuan Yang, Ming-Ming Cheng, Junwei Han, and Philip Torr. Large-scale unsupervised semantic segmentation. IEEE Transactions on Pattern Analysis and Machine Intelligence, 2022.

---

> > ### Comment · Reviewer_BFe6 · 2024-08-12
> >
> > Thanks for the authors' response. I would like to confirm with you the details of your experiment settings. Specifically, I wonder how do you calculate the attention map when using text-guided attention and vision-only attention in this experiment (and how in Fig. 1).

---

> > > ### Author Response · Authors · 2024-08-12
> > >
> > > Thanks for your question. We will incorporate these discussions into the final version.
> > >
> > > We compute the **text-guided attention** $A(x)$ by simply multiplying the text embedding $g(t)$ with the vision embedding $f_{g}(x)$. Specifically, this is expressed as $A(x)=f_{g}(x) \cdot g(t)^ \mathsf{T} $, where  $f_{g}(x)$ represents the global image features before the pooling operation of the class token $f(x)$ (as described in Equation 5 and in Lines 162-166 of the paper). Text-guided attention incorporates textual context, potentially offering more precise and context-aware localization of relevant image features.
> > >
> > > For comparison, the **vision-only attention** is obtained using the widely adopted visualization technique in deep learning, Grad-CAM [49]. The Grad-CAM heatmap $L_{\text{Grad-CAM}}^c $ for class $c$ is defined as $L_{\text{Grad-CAM}}^c = \text{ReLU} \left( \sum_k \alpha_k^c A^k \right)$, where $\alpha_k^c = \frac{1}{Z} \sum_i \sum_j \frac{\partial y^c}{\partial A_{ij}^k}$. Here, $y^c$ denotes the class score, $i,j$ are the spatial dimensions within the activation map of the $k$-th convolutional layer $A^k$, and $Z$ represents the total number of spatial locations $i \times j$ in the feature map. The Grad-CAM method highlights regions of the image that are most relevant to the class prediction by calculating the gradients of the class score with respect to the activation map.
> > >
> > > Figure 1 demonstrates the results of our text-guided attention. We first convert a grayscale attention map into a color map using the ‘cv2.applyColorMap’ function. This colorized attention map is then overlaid onto the original image through a weighted sum: $ image \times 0.4 + attention × 0.6$ (these weights can be adjusted to achieve the desired visual emphasis), as similarly demonstrated in references [e] and [f]. The resulting image, shown in Figure 1, effectively highlights the areas of focus for the model, providing a clear visualization of its attention.
> > >
> > >
> > > [49] Ramprasaath R Selvaraju, Michael Cogswell, Abhishek Das, Ramakrishna Vedantam, Devi Parikh, and456
> > > Dhruv Batra. Grad-cam: Visual explanations from deep networks via gradient-based localization. In457
> > > Proceedings of the IEEE international conference on computer vision, pages 618–626, 2017.
> > >
> > > [e] Song, Y., Jang, S., Katabi, D., & Son, J. (2023). Unsupervised Object Localization with Representer Point Selection. 2023 IEEE/CVF International Conference on Computer Vision (ICCV), 6511-6521.
> > >
> > > [f] Li, Y., Wang, H., Duan, Y., & Li, X. (2023). CLIP Surgery for Better Explainability with Enhancement in Open-Vocabulary Tasks. ArXiv, abs/2304.05653.

---

> > > > ### Comment · Reviewer_BFe6 · 2024-08-12
> > > >
> > > > Thanks for the authors' response and their valuable insights. My concerns have been addressed. I wil increase my rating to Weak Accept.

---

### Official Review · Reviewer_qndz · 2024-07-10

**Soundness:** 3
**Presentation:** 2
**Contribution:** 2
**Rating:** 5
**Confidence:** 3

**Summary:**

This paper proposes an approach to improve the adversarial robustness of vision-language models while maintaining performance on clean images. The key idea is to aligns the attention maps of adversarial examples with clean examples. Extensive experiments demonstrate the effectiveness of the proposed method in zero-shot adversarial robustness across multiple datasets while maintaining high clean accuracy.

**Strengths:**

1. The proposed method is simple, straightforward and effective. It leverages text-guided attention to significantly enhance zero-shot adversarial robustness while maintaining clean accuracy across diverse datasets.

2. The experiments are comprehensive. The results on 16 datasets demonstrate consistent improvements over baseline methods.

**Weaknesses:**

1. The method is only demonstrated on CLIP, raising questions about its applicability to other vision-language models or architectures.

2. The method introduces additional hyperparameters without a clear strategy for tuning them across different datasets or tasks.

3. No error bars or statistical significance tests reported for the experimental results.

**Questions:**

1. How sensitive is the method to the choice of hyperparameters? Is there a principled way to select these across different datasets or tasks?

2. Are there any scenarios or types of adversarial attacks where this method performs poorly? What are the limitations of this approach?

**Limitations:**

The authors did not explicitly address the limitations such as the types of adversarial attacks where this method performs poorly.

---

> ### Author Rebuttal · Authors · 2024-08-06
>
> **Q1: Applicability to other vision-language models.**
>
> **A1**: We follow TeCoA and PMG-AFT, focusing on improving the zero-shot adversarial robustness of the CLIP model for classification tasks. To further validate the effectiveness of our method as suggested by the reviewer, we replaced the CLIP model with another vision-language model, **OpenFlamingo-3B**[a]. In this setup, ViT-L/14 serves as the vision encoder and MPT-1B as the language encoder. Additionally, we evaluated our method on two other tasks: image captioning and visual question answering (VQA). We report the CIDEr score for image captioning and VQA accuracy for visual question answering tasks. We employ the APGD attack[b] with a strength of epsilon 8/255 for 10 iterations. The results are shown below. Our method outperforms FARE in most scenarios for both image captioning and VQA tasks across a range of datasets.  We believe that with task-specific design enhancements, our results can be further improved.
>
> Table 2: CIDEr Scores for Image Captioning Task with OpenFlamingo-3B.
>
> |        |       | COCO |   Flickr30k 	|
> |---------|-------|------------|-----------|
> | Robust | FARE |  	3.68  |   	2.71    	|
> |        | Ours  |  	**4.13** |   	**2.90**    	|
> |  Clean | FARE |  	3.09  |   	3.02    	|
> |        | Ours  |  	**3.56**  |   **3.13**   	|
>
> Table 3: Accuracy for Visual Question Answering (VQA) with OpenFlamingo-3B.
>
> |        |       	| TextVQA |  VQAv2 	|
> |---------|-------|------------|-----------|
> | Robust | FARE |  	 3.58 |     31.88   	|
> |        | Ours  |    **4.44** |   **32.14**   	|
> |  Clean | FARE |  	  3.40  |  	**35.38**   	|
> |        | Ours  |  	  **4.38** |   	34.72 	|
>
> [a] Awadalla, A., et al. OpenFlamingo: an opensource framework for training large autoregressive vision-language models. arXiv:2308.01390, 2023.
>
> [b] Croce, F. and Hein, M. Reliable evaluation of adversarial robustness with an ensemble of diverse parameter-free attacks. In ICML, 2020.
>
> **Q2: Strategy for tuning hyper-parameters.**
>
> **A2:** Following the protocol of previous works (TeCoA,PMG-AFT,FARE), we fine-tuned the CLIP model on adversarial samples from a single dataset (Tiny-ImageNet in our case) for `adversarial fine-tuning' and subsequently evaluated its performance across 15 datasets, including Tiny-ImageNet itself. Thus we only need to tune hyperparameters on just one training dataset. We randomly selected 80% of the training set for training and the remaining 20% for validation to choose the hyperparameters. The validation set results are shown below. The final results on the test set were obtained by training on the entire training set using the optimal hyperparameters (alpha=0.08, bata=0.05) identified from the validation set.
>
> Table 5 : Results on validation set of Tiny-ImageNet dataset.
>
> | Hyper-parameters | Robust | Clean | Average |
> |-------|-------|-------|-------|
> |   alpha=0.07 beta=0.05  |  64.32  |  75.92  |  70.12  |
> |   alpha=0.08 beta=0.04  |  47.25  |  76.20  |  61.72  |
> |   alpha=0.08 beta=0.05  |  64.01  |	  77.79	 |  **70.90** |
> |   alpha=0.08 beta=0.06  |  58.28  |  76.08  |  67.18  |
> |   alpha=0.09 beta=0.05  |  46.20  |  76.10  |  61.15  |
>
> **Q3: Reporting error bars or statistical significance tests.**
>
> **A3**: Thank you for your suggestions. We acknowledge the absence of reporting error bars and statistical significance tests in the initial submission. To address this, we have included the standard deviation  from three runs of the main experiments to demonstrate the stability of our method. We will include comprehensive statistical significance tests in the subsequent versions of the paper.
>
> Table 6: Results with standard deviation  from three runs across 16 datasets.
>
> |  Test  | Tiny- |  C.10  |  C.100 |   STL  |   SUN   |   Food  | Pets | Flowers |   |
> |------|-------------|----------|----------|----------|----------|----------|----------|----------|----------|
> | Robust |   63.95±0.11  | 61.45±0.67 | 35.27±0.07 | 84.22±0.21 | 33.22±0.39 | 33.97±0.20 | 57.75±0.76 | 34.55±0.35 |  |
> |  Clean |   75.72±0.12  | 86.46±0.26 | 56.52±0.35 | 93.48±0.19 | 51.99±0.25 | 57.59±0.34 | 77.32±0.30 | 48.08±0.37 |  |
>
> | | DTD    |   EuroS.  | Airc.|  ImageN.  | Ca.101 | Ca.256 | Cars |    PCAM    |   Avg.  |
> |------|-------------|----------|----------|----------|----------|----------|----------|----------|----------|
> | Robust |22.08±0.16 | 14.27±0.26 |   4.75±0.27   | 28.74±0.11 |  70.97±0.42 |  60.06±0.46 |  20.40±0.68  | 47.76±0.35 | 42.09±0.12 |
> |  Clean | 29.06±0.35 | 24.24±0.49 |   11.93±0.27  | 48.04±0.06 |  80.70±0.29 |  74.74±0.18 |  36.62±1.03  | 49.58±0.17 | 56.44±0.08 |
>
> **Q4: Adversarial attack scenarios where this method performs poorly and its limitations.**
>
> **A4**: In this paper, we primarily train using the PGD attack and validate our approach on both PGD and AutoAttack. Our method significantly outperforms other methods on the PGD attack and achieves comparable results with state-of-the-art methods on AutoAttack, which integrates multiple attack strategies (APGD_CE, APDG_DLR, FAB, Square Attack). These results suggest that our method is robust but could be further enhanced to withstand stronger and more complex attacks. One potential improvement is to design a more robust text-guided attention mechanism.
>
> **We will revise and add the corresponding context in the final version.**

---

> ### Comment · Reviewer_qndz · 2024-08-12
>
> I thank the authors for their responses. All my concerns have been addressed. I will increase my rating from 4 to 5.

---

### Official Review · Reviewer_w4iU · 2024-07-11

**Soundness:** 3
**Presentation:** 3
**Contribution:** 3
**Rating:** 6
**Confidence:** 2

**Summary:**

The paper proposes a framework, Text-Guided Attention for Zero-Shot Robustness (TGA-ZSR), to enhance the robustness of vision-language models (VLMs) against adversarial attacks. The proposed method incorporates two modules: the Attention Refinement module and the Attention-based Model Constraint module. The Attention Refinement module aligns the text-guided attention of adversarial examples with clean examples, while the Attention-based Model Constraint module maintains the model’s performance on clean samples.

**Strengths:**

1. The use of text-guided attention allows to enhance zero-shot robustness.
2. Extensive evaluation of the approach was conducted across diverse datasets.
3. The proposed method outperforms current state-of-the-art techniques in both zero-shot robust accuracy.
4. The paper provides a thorough analysis of the impact of adversarial attacks on text-guided attention, offering insights into the model's decision-making process.
5. The method improves robustness without significantly sacrificing performance on clean data, achieving a favorable trade-off.

**Weaknesses:**

1. The incorporation of text-guided attention mechanisms and multiple modules may increase the complexity of implementation and computational overhead.
2. Although the authors claim zero-shot robustness in Vision-Language Models, the experiments and methods only target the CLIP model.
3. Missing limitation in Section 5.

**Questions:**

1. What is the additional computational overhead introduced by the Attention Refinement and Attention-based Model Constraint modules? How does this impact training and inference times compared to baseline methods?
2. How well does the proposed method generalize to other types of vision-language tasks beyond image classification, such as image captioning or visual question answering?
3. Why is the proposed method fine-tuned on Tiny-ImageNet rather than ImageNet? Both TeCoA [1] and FARE [2] are trained on ImageNet.

[1] Mao, C., Geng, S., Yang, J., Wang, X., & Vondrick, C. (2022). Understanding zero-shot adversarial robustness for large-scale models.

[2] Schlarmann, C., Singh, N. D., Croce, F., & Hein, M. (2024). Robust clip: Unsupervised adversarial fine-tuning of vision embeddings for robust large vision-language models.

---

> ### Author Rebuttal · Authors · 2024-08-06
>
> **Q1: Discussion of additional computational overhead and training/inference time.**
>
> **A1**:Thank you for your suggestion. We have evaluated our method against others in terms of memory usage, training time, and test time, and the findings are summarized below:
>
> **Memory Usage**: Our method increases memory consumption by approximately 15% compared to the state-of-the-art method PMG-AFT. This is due to the additional computation required for the text-guided attention map.
>
> **Training Time**: The training time for our method is comparable to that of PMG-AFT, which utilizes a KL divergence constraint on logits.
>
> **Test Time**: The test time remains consistent across all methods.
>
> Table 1: Comparison of memory usage, training time, and test time.
>
> | Method  | Train memory usage	|Train time (per epoch/batch) | Test time (per batch) |
> |---------|-----------------------------------|-----------|-----------|
> | CLIP    	|	0Mb 	 |   	0s / 0s                     | 21s       |
> | TeCoA   	| 	12873Mb | 512s / 0.65s                   | 21s       |
> | PMG-AFT 	| 	18449Mb |	828s / 1.06s                  | 21s       |
> | Ours    	|	21227Mb | 	885s /1.13s                    | 21s       |
>
> **Q2:  Applying to other types of vision-language tasks on other tasks.**
>
> **A2**: We follow TeCoA and PMG-AFT, focusing on improving the zero-shot adversarial robustness of the CLIP model for classification tasks. To further validate the effectiveness of our method as suggested by the reviewer, we replaced the CLIP model with another vision-language model, **OpenFlamingo-3B**[a]. In this setup, ViT-L/14 serves as the vision encoder and MPT-1B as the language encoder. Additionally, we evaluated our method on two other tasks: image captioning and visual question answering (VQA). We report the CIDEr score for image captioning and VQA accuracy for visual question answering tasks. We employ the APGD attack[b] with a strength of epsilon 8/255 for 10 iterations. The results are shown below. Our method outperforms FARE in most scenarios for both image captioning and VQA tasks across a range of datasets.  We believe that with task-specific design enhancements, our results can be further improved.
>
> Table 2: CIDEr Scores for Image Captioning Task with OpenFlamingo-3B.
>
> |        |       | COCO |   Flickr30k 	|
> |---------|-------|------------|-----------|
> | Robust | FARE |  	3.68  |   	2.71    	|
> |        | Ours  |  	**4.13** |   	**2.90**    	|
> |  Clean | FARE |  	3.09  |   	3.02    	|
> |        | Ours  |  	**3.56**  |   **3.13**   	|
>
> Table 3: Accuracy for Visual Question Answering (VQA) with OpenFlamingo-3B.
>
> |        |       	| TextVQA |  VQAv2 	|
> |---------|-------|------------|-----------|
> | Robust | FARE |  	 3.58 |     31.88   	|
> |        | Ours  |    **4.44** |   **32.14**   	|
> |  Clean | FARE |  	  3.40  |  	**35.38**   	|
> |        | Ours  |  	  **4.38** |   	34.72 	|
>
> [a] Awadalla, A., et al. OpenFlamingo: an opensource framework for training large autoregressive vision-language models. arXiv:2308.01390, 2023.
>
> [b] Croce, F. and Hein, M. Reliable evaluation of adversarial robustness with an ensemble of diverse parameter-free attacks. In ICML, 2020.
>
> **Q3: Missing limitation.**
>
> **A3**: We mentioned the limitation in the final sentence of Section 5. We will emphasize and expand upon it in future versions.
>
> **Q4: Train on ImageNet dataset.**
>
> **A4**: We follow the state-of-the-art method PMG-AFT, which was fine-tuned on Tiny-ImageNet. Thank you for your suggestion. Due to time constraints, we further evaluate our method on the ImageNet_subset (a random selection of 100 classes from the full ImageNet dataset). The results are shown below:
>
> Table 4: Zero-shot adversarial robust accuracy and clean accuracy across 16 datasets by training on ImageNet_subset.
>
> | Test | Robust | Clean | Average |
> |-------|-------|-------|-------|
> | CLIP | 4.90 | 64.42 | 34.66 |
> | TeCoA | 20.42 | 40.68 | 30.55 |
> | FARE | 11.41 | 60.00 | 35.70 |
> | PMG-AFT | 23.93 | 43.10 | 33.51 |
> | Ours | 24.74 | 46.90 | **35.82** |
>
> **We will revise and add the corresponding context in the final version.**

---

> > ### Comment · Reviewer_w4iU · 2024-08-11
> >
> > Thank you for your detailed response. I will be maintaining my initial score.

---

> > > ### Author Response · Authors · 2024-08-11
> > >
> > > Thank you for taking the time to review our response and for your support. We are glad to hear that the response addressed your concerns.

---

### Author Rebuttal · Authors · 2024-08-07

**1. Summary**: We thank the reviewers for their positive and constructive comments. The reviewers agree that the topic is interesting and that the proposed method is novel, simple, and effective. All reviewers appreciate the comprehensive, solid, thorough, and detailed experiments. They also acknowledge that the paper is well-written and organized. The detailed feedback is as follows:

| Reviewers                 | Aspects        | Comments                                                                               |
|---------------------------|----------------|----------------------------------------------------------------------------------------|
| BFe6                      | Topic          | Interesting                                                                            |
| mEBo & qndz & BFe6        | Method         | Novel; simple, straightforward and effective; simple but intuitive and effective       |
| w4iU & qndz & BFe6 & mEBo | Experiments    | Thorough analysis, offering insights; comprehensive; solid; detailed ablation analysis |
| BFe6 & mEBo               | Representation | Well written; well written and organized, clear descriptions of motivation and method |

The reviewers' major comments suggest that additional analysis could provide a deeper understanding of the proposed method. Specifically, they recommended including: Analysis of additional computational overhead / Application to other types of vision-language models / Results on more attacks / Discussion of scenarios where our method performs poorly. **Note that the primary analyses widely used in existing zero-shot adversarial robustness methods for VLMs have already been provided in the paper and supplementary materials**. The additional analyses suggested by the reviewers are complementary and would enhance the understanding of our proposed method.

**2. More analysis of the proposed method**:

- *Additional computational overhead and training/inference time of the models are shown in Table 1.*

Table 1: Comparison of memory usage, training time, and test time.

| Method  | Train memory usage	|Train time (per epoch/batch) | Test time (per batch) |
|---------|-----------------------------------|-----------|-----------|
| CLIP    	|	0Mb 	 |   	0s / 0s                     | 21s       |
| TeCoA   	| 	12873Mb | 512s / 0.65s                   | 21s       |
| PMG-AFT 	| 	18449Mb |	828s / 1.06s                  | 21s       |
| Ours    	|	21227Mb | 	885s /1.13s                    | 21s       |

- *The results with OpenFlamingo-3B on image captioning and VQA are shown in Table 2 and 3.*

Table 2: CIDEr Scores for Image Captioning Task with OpenFlamingo-3B.

|        |       | COCO |   Flickr30k 	|
|---------|-------|------------|-----------|
| Robust | FARE |  	3.68  |   	2.71    	|
|        | Ours  |  	**4.13** |   	**2.90**    	|
|  Clean | FARE |  	3.09  |   	3.02    	|
|        | Ours  |  	**3.56**  |   **3.13**   	|

Table 3: Accuracy for Visual Question Answering (VQA) with OpenFlamingo-3B.

|        |       	| TextVQA |  VQAv2 	|
|---------|-------|------------|-----------|
| Robust | FARE |  	 3.58 |     31.88   	|
|        | Ours  |    **4.44** |   **32.14**   	|
|  Clean | FARE |  	  3.40  |  	**35.38**   	|
|        | Ours  |  	  **4.38** |   	34.72 	|

- *The results training on ImageNet_subset are shown in Table 4.*

Table 4: Zero-shot adversarial robust accuracy and clean accuracy across 16 datasets by training on ImageNet_subset.

| Test | Robust | Clean | Average |
|-------|-------|-------|-------|
| CLIP | 4.90 | 64.42 | 34.66 |
| TeCoA | 20.42 | 40.68 | 30.55 |
| FARE | 11.41 | 60.00 | 35.70 |
| PMG-AFT | 23.93 | 43.10 | 33.51 |
| Ours | 24.74 | 46.90 | **35.82** |

- *The results on the validation set of Tiny-ImageNet are shown in Table 5.*

Table 5 : Results on validation set of Tiny-ImageNet dataset.

| Hyper-parameters | Robust | Clean | Average |
|-------|-------|-------|-------|
|   alpha=0.07 beta=0.05  |  64.32  |  75.92  |  70.12  |
|   alpha=0.08 beta=0.04  |  47.25  |  76.20  |  61.72  |
|   alpha=0.08 beta=0.05  |  64.01  |  77.79. |  **70.90** |
|   alpha=0.08 beta=0.06  |  58.28  |  76.08  |  67.18  |
|   alpha=0.09 beta=0.05  |  46.20  |  76.10  |  61.15  |

- *The results included the standard deviation  from three runs of the main experiments are shown in Table 6.*

(Table 6 is available in the response to Reviewer qndz due to character limit constraints and is not shown here.)

- *Results with CW attack are shown in Table 7.*

Table 7: Zero-shot adversarial robust accuracy and clean accuracy across 16 datasets  with **CW attack**.

|   |Methods |  Avg.  |  |Methods |  Avg.  |
|:------:|:-------:|:-------------:|:-----:|:-------:|:-------------:|
|        |   CLIP  |       3.64  |        |   CLIP  |      64.42  |
| **Robust** | PMG-AFT |    31.07 |    **Clean** | PMG-AFT |    46.58  |
|        |   Ours  |     40.50  |         |   Ours  |    55.72  |

**3. More explanation of the method.**

- *Results on AutoAttack*: In this paper, we primarily train using the PGD attack and validate our approach on both PGD and AutoAttack. Our method significantly outperforms other methods on the PGD attack and achieves comparable results with state-of-the-art methods on AutoAttack, which integrates multiple attack strategies. These results suggest that our method is robust but could be further enhanced to withstand stronger and more complex attacks.

- *How to use class token*: In Figure 2, the class token used as  $f(x_a)$ is obtained from $f_g^{tar}(x_a)$ after pooling. We will clarify this in more detail in future versions.

- *"Generalizability" and "adversarial robustness"*: Our goal is to maintain the generalization and enhance the adversarial robustness of the original CLIP model. We will modify it in the revised version of the paper.

---

### Decision · Program_Chairs · 2024-09-25

**Decision:**

Accept (poster)

**Comment:**

The paper introduces a method to improve the robustness of visual language pretrained models. Building on insights from text-guided attention, the authors propose two modules—the Attention Refinement module and the Attention-based Model Constraint module—to enhance model robustness. While the reviewers raised concerns about additional comparisons, complexities, and unclear wording, the AC found that the rebuttal effectively addressed all these issues. In light of this, the AC recommends acceptance. The authors are encouraged to incorporate all changes and additional experimental results into the final camera-ready version of the paper.